# Manifolds, Random Matrices and Spectral Gaps: The geometric phases of generative diffusion

**Enrico Ventura**[1,*]          **Beatrice Achilli**[1,*]          **Gianluigi Silvestri**[2,3,*]

**Carlo Lucibello**[1]                                              **Luca Ambrogioni**[2]

[1]Department of Computing Sciences, BIDSA, Bocconi University, Milan, MI 20100, Italy.
[2]Donders Institute for Brain, Cognition and Behaviour, Radboud University,
6500 HD Nijmegen, the Netherlands.
[3]OnePlanet Research Center, imec-the Netherlands, Wageningen, the Netherlands.

## Abstract

In this paper, we investigate the latent geometry of generative diffusion models under the manifold hypothesis. For this purpose, we analyze the spectrum of eigenvalues (and singular values) of the Jacobian of the score function, whose discontinuities (gaps) reveal the presence and dimensionality of distinct sub-manifolds. Using a statistical physics approach, we derive the spectral distributions and formulas for the spectral gaps under several distributional assumptions, and we compare these theoretical predictions with the spectra estimated from trained networks. Our analysis reveals the existence of three distinct qualitative phases during the generative process: a trivial phase; a manifold coverage phase where the diffusion process fits the distribution internal to the manifold; a consolidation phase where the score becomes orthogonal to the manifold and all particles are projected on the support of the data. This 'division of labor' between different timescales provides an elegant explanation of why generative diffusion models are not affected by the manifold overfitting phenomenon that plagues likelihood-based models, since the internal distribution and the manifold geometry are produced at different time points during generation.

## 1 Introduction

Generative diffusion models have revolutionized the fields of computer vision and generative modeling, achieving state-of-the-art performance on image generation (Ho et al., 2020; Song and Ermon, 2019; Yang et al., 2021) and video generation (Ho et al., 2022; Singer et al., 2022; Blattmann et al., 2023; Tim et al., 2024). Generative diffusion models synthesize images through a stochastic dynamical denoising process. Experimental and theoretical arguments suggest that different features such as frequency modes and class labels are generated at different times during the process. For example, it has been shown that separation between isolated classes, as in the case of mixture of Gaussian models, happens at critical phase transition points of spontaneous symmetry breaking (speciation events) (Biroli et al., 2024). It is also well known that subspaces corresponding to different frequency modes emerge at different times of diffusion (Kingma and Gao, 2024). This idea has been recently refined by (Kadkhodaie et al., 2024), who showed that diffusion models give rise to a local decomposition of the image manifold into a basis of geometry-adaptive harmonic basis functions. These decomposition phenomena cannot be directly explained in terms of critical phase transitions

---

*These authors contributed equally to this work

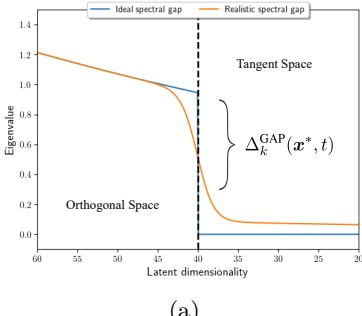
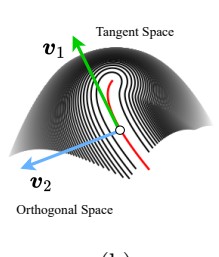
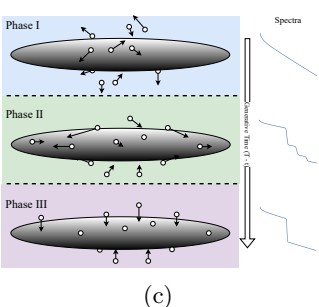

(a)          (b)          (c)

Figure 1: (a) Visualization of the gaps in the spectrum of the (negative) Jacobian of the score for data supported on a latent manifold. Blue line: idealized spectrum of distribution with uniform internal density; Orange line: spectrum of a more realistic distribution. (b) Sketch of the local structure of data-manifold with tangent and orthogonal components of the score function. (c) Sketch of the geometric phases of generative diffusion and their trace measurable from the eigenspectrum.

as they are fundamentally linear processes. In this paper, we will provide a precise theoretical analysis of the separation of subspaces for data defined on low dimensional linear manifolds.

Our main contributions are: I) an in-depth theoretical random-matrix analysis of the distribution of Jacobian spectra in diffusion models on linear manifolds and II) a detailed experimental analysis of Jacobian spectra extracted from trained networks on linear manifolds and on image datasets. The analysis of these spectra is important as it provides a detailed picture of the latent geometry that guides the generative diffusion process. We show that the linear theory predicts several phenomena that we observed in trained networks. Based on our result, we divide the generative process into three qualitatively different phases: **trivial phase**, **manifold coverage phase** and **manifold consolidation phase**. Using these concepts, we provide a concise explanation of why diffusion models can avoid the *manifold overfitting* pathology that characterizes likelihood-based generative models (Loaiza-Ganem et al., 2022).

## 2    THE MANIFOLD HYPOTHESIS

The manifold hypothesis states that the distribution on natural data, such as images and sound recordings, is (approximately) supported on a $m-$dimensional manifold $\mathcal{M}$ embedded in a larger euclidean ambient space $\mathbb{R}^d$ (Peyré, 2009; Fefferman et al., 2016). While a data probability distribution supported on a $m < d$ manifold $\mathcal{M}$ cannot be expressed using a proper density function, we loosely define such density as

$$p_0(\boldsymbol{x}) = \delta_{\mathcal{M}}(\boldsymbol{x})\, \rho(\boldsymbol{x})\ , \tag{1}$$

where $\delta_{\mathcal{M}}$ is the Dirac function for the manifold and such that $\int_{\mathbb{R}^d} \delta_{\mathcal{M}}(\boldsymbol{x}) \bullet d\boldsymbol{x} = \int_{\mathcal{M}} \bullet\, d\boldsymbol{x}$. We call $\rho(\boldsymbol{x})$ the *internal density*, that is the density restricted to the manifold. The density $p_0(\boldsymbol{x})$ is zero outside the manifold and diverges on the manifold. The hidden manifold model proposed by Goldt et al. (2020) is able to reproduce $N$ data $\boldsymbol{y}^\mu$ that are embedded in a latent $m$-dimensional space, as

$$\boldsymbol{y}^\mu = \Phi\left(F \boldsymbol{z}^\mu\right), \tag{2}$$

where $\Phi(\boldsymbol{x})$ is a non-linear function of the $d$-dimensional vector $\boldsymbol{x}$, $F \in \mathbb{R}^{d \times m}$ is a rectangular matrix that projects latent vectors $\boldsymbol{z}^\mu$ on the manifold. In this work we are going to study generative diffusion while aligning with this modeling frame.

## 3    BACKGROUND ON GENERATIVE DIFFUSION MODELS

Here, we will consider a simple variance-exploding forward process where the data $\boldsymbol{x}_0 \sim p_0(\boldsymbol{x})$ evolves according to the equation

$$d\boldsymbol{x}_t = d\boldsymbol{Z}_t \tag{3}$$

where $d\boldsymbol{Z}_t$ is a standard Brownian motion. The formal solution of Eq. (3) can be given in terms of the heat kernel $p_t(\boldsymbol{x}_t) = \mathbb{E}_{\boldsymbol{x}_0 \sim p_0}\left[\frac{1}{\sqrt{2\pi t}}e^{-\frac{\|\boldsymbol{x}_t - \boldsymbol{x}_0\|_2^2}{2t}}\right]$. The *target distribution* $p_0(\boldsymbol{x})$ is then recovered by reversing the diffusion process (Anderson, 1982). We initialize this reverse process from $\boldsymbol{x}_{t_f} \sim \mathcal{N}\left(0, t_f I_d\right)$ at some large time $t_f$, which evolves backward in time according to

$$\mathrm{d}\boldsymbol{x}_t = -\nabla_{\boldsymbol{x}} \log p_t(\boldsymbol{x}_t)\mathrm{d}t + d\boldsymbol{Z}_t \tag{4}$$

The function $s(\boldsymbol{x}, t) = \nabla_{\boldsymbol{x}} \log p_t(\boldsymbol{x})$ is the so-called score function. From a set of training points $\{\boldsymbol{y}^1, \ldots, \boldsymbol{y}^N\} \overset{\text{iid}}{\sim} p_0$, we can train a neural approximation of $s(\boldsymbol{x}, t)$ by learning a denoising autoencoder $\hat{\boldsymbol{\epsilon}}_{\boldsymbol{\theta}}(\mathbf{x}, t)$, which is trained to recover the standardized noise $\epsilon_t$ from the noisy state $x_t = x_0 + \sqrt{t}\epsilon_t$ (Hyvärinen and Dayan, 2005; Vincent, 2011; Ho et al., 2020). In order to avoid singularities in neural network output, the learned score is parametrized as $\hat{s}_{\boldsymbol{\theta}}(\boldsymbol{x}, t) = -\frac{\hat{\boldsymbol{\epsilon}}_{\boldsymbol{\theta}}(\boldsymbol{x},t)}{\sqrt{t}}$ .

## 4    Dynamic latent manifolds and spectral gaps

Consider a generative diffusion model with $p_0(\boldsymbol{x})$ defined on a $d$-dimensional manifold $\mathcal{M}$ according to Eq. (1). In the course of the diffusion process, we can define a time-dependent locus of points

$$\mathcal{M}_t = \{\boldsymbol{x}^* \mid \tilde{s}_{\mathcal{M}}(\boldsymbol{x}^*, t) = 0, \text{ with } J_{\mathcal{M}}(\boldsymbol{x}^*, t) \text{ n.s.d.}\} , \tag{5}$$

that we name **stable latent set** of the process. In Eq. 5 we have used $\mathcal{M} \equiv \mathcal{M}_0$. The negative semi-definiteness (n.s.d.) is a stability condition on the Jacobian matrix $J_{\mathcal{M}}(\boldsymbol{x}, t)$ of the support score $\tilde{s}_{\mathcal{M}}(\boldsymbol{x}^*, t)$, defined as the score function obtained from the uniform data distribution $\tilde{p}_0(\boldsymbol{x}) = \frac{1}{|\mathcal{M}|}\delta_{\mathcal{M}}(\boldsymbol{x})$. Due to the noise, the diffusing particles typically explore shells of a radius that concentrates on $\sqrt{t}$ around each point of the latent stable set. For a small perturbation $\boldsymbol{p}$ around a point $\boldsymbol{x}^*$ on the latent manifold at time $t$, the score function is well approximated by its linearization:

$$s(\boldsymbol{p}, t) \approx J(\boldsymbol{x}^*, t)\, \boldsymbol{p} = -\sum_j \left(\boldsymbol{v}_j \cdot \boldsymbol{p}\right) \lambda_j(\boldsymbol{x}^*, t)\boldsymbol{v}_j , \tag{6}$$

where $J(\boldsymbol{x}^*, t)$ is the Jacobian of the score and the $\boldsymbol{v}_j$ and $\lambda_j(\boldsymbol{x}^*, t)$ are respectively the $j$-th eigenvector and the associated eigenvalue of $-J(\boldsymbol{x}^*, t)$. The spectrum of eigenvalues provides detailed information concerning the local geometry of stable latent set. Perturbations aligned with the tangent space of $\mathcal{M}_t$ correspond to small eigenvalues, while orthogonal perturbations correspond to high eigenvalues, as the score tends to push the stochastic dynamics towards its fixed-points. Therefore, we can estimate the dimensionality of the manifold from the location of a drop (i.e. a sharp change) in the sorted spectrum of eigenvalues (Stanczuk et al., 2022). This is visualized in Fig. 1, panels (a) and (b). This drop corresponds exactly to a gap (i.e. a separation) in the eigenvalues spectrum; in the following, we will refer to both as gaps.

### 4.1    Subspaces and intermediate gaps

Consider the situation where the internal density $\rho_{\text{int}}(\boldsymbol{x})$ is not locally flat around a point $\boldsymbol{x}^* \in \mathcal{M}$. In this case, at a finite time $t$ the actual score function does not vanish on the latent stable set $\mathcal{M}_t$ as there is a gradient of the log-density along the tangent directions. This implies that the spectrum of tangent eigenvalues can have a series of sub-gaps with separate different tangent subspaces with different 'local variance'. In image generation tasks, these subspaces are often associated with different frequency modes, as noted in (Kingma and Gao, 2024). Consequently, we can quantify the sensitivity to the internal density at time $t$ by studying the statistics and temporal evolution of intermediate gaps

$$\Delta_k^{\text{GAP}}(\boldsymbol{x}^*, t) = \lambda_{k+1}(\boldsymbol{x}^*, t) - \lambda_k(\boldsymbol{x}^*, t) , \tag{7}$$

where the indices $k$ depend on the dimensionality of the subspaces. Note however that under realistic data distributions it is unlikely to find sharp intermediate discontinuities since each subspace will have a different eigenvalue, resulting in a smooth gradient.

## 5   Phenomenology of generative diffusion on manifolds

This section contains an intuitive picture that follows from our theoretical results on linear models, which we will fully outline in the next section. The theory considers the case of a linear manifold with Gaussian internal distribution. A linear-manifold data-model is made of a set of points

$$\boldsymbol{y}^{\mu} = F\boldsymbol{z}^{\mu}, \tag{8}$$

where $F$ and $\boldsymbol{z}^{\mu}$ have been introduced in Section 2. While only linear models are theoretically tractable, we conjecture that their phenomenology captures the main features of subspace separation in the tangent space of curved manifolds (see Supp. D for further details). We validated the theory using networks trained on both linear data and highly non-linear data such as natural images (see Sections 7 and 8). Based on the dynamics of the spectral gaps, we found that the generative dynamics of $\boldsymbol{x}_t$ according to Eq. (4) can be separated into three distinct phases. The phase separation does not correspond to singularities as there are cross-over events, not genuine phase transitions. During all our analysis we will exclusively work with *eigenvalues* since, in the linear manifold model, the Jacobian of the true score is symmetric. The same phenomenology is nevertheless fully appreciable when using the *singular values* in our experiments with neural approximations of the score.

### 5.1   Phase I: The trivial phase

In the **trivial phase**, the diffusing particle moves according to the noise distribution without strong biases towards the manifold directions. In this dynamic regime, the stable latent set $\mathcal{M}_t$ is a single point surrounded by an isotropic quadratic well of potential. The spectral gaps are not visible and all eigenvalues have approximately the same value due to the isotropy of the noise distribution. This trivial phase is analogous to the initial phases described in (Raya and Ambrogioni, 2024) and (Biroli et al., 2024).

### 5.2   Phase II: Manifold coverage

The **manifold coverage phase** begins with the opening of the first of a series of spectral gaps corresponding to local subspaces. In this phase, different subspaces with different variances can therefore be identified by intermediate gaps in the spectra, as sketched in Fig. 1, panel (c). When the intermediate gaps are opened, the diffusing particles spread across the manifold directions according to their relative variances. In other words, during this regime of generative diffusion, the process fits the distribution of the data internal to the manifold.

We assume low-rank covariance $\Sigma = FF^{\top}$ for the data distribution. In terms of random matrix theory, the gap-forming phenomenology has two distinct processes: the emergence of intermediate gaps (i.e. steps in the dimensionality plot) between separated bulks of the spectrum, and the opening of a final gap that allows to infer the dimensionality of the full manifold. Our analysis gives us the time scale at which such intermediate gaps are maximally opened, i.e.

$$t_{\max}^{(k)} = \sqrt{\gamma_+(\sigma_k)\,\gamma_-(\sigma_{k+1})}, \tag{9}$$

where $\gamma_-(\sigma_{k+1})$ and $\gamma_+(\sigma_k)$ are specific eigenvalues of $\Sigma$ (see Fig. 8) associated with two hierarchically consecutive variances (see Supp. A.2 for an exhaustive analysis). In most cases, when $\sigma_{k+1}^2 \ll \sigma_k^2$, the dependence on the two variances is $\mathcal{O}(\sigma_k \cdot \sigma_{k+1})$. This is the timescale where the score is maximally sensitive to the relative variance of the two subspaces, which guides the particles toward the correct internal distribution.

### 5.3   Phase III: Manifold consolidation

Finally, the **manifold consolidation phase** is characterized by the asymptotic closure of the intermediate gaps and the sharpening of the total manifold gap, indicating the full dimensionality of $\mathcal{M}$. In this final regime, the score assumes the form

$$\nabla_{\mathbf{x}} \log p_t(\mathbf{x}) \simeq \frac{1}{t}\Big[\Pi - I_d\Big]\mathbf{x}. \tag{10}$$

where $\Pi = F(F^\top F)^{-1} F^\top$ is the projection matrix over the manifold. The component of the score orthogonal to the manifold diverges proportionally to $t^{-1}$, while the tangent components converge to a constant and become therefore negligible in this regime. This results in the consolidation of the gap corresponding to the manifold dimensionality $m$ and to the (relative) closure of the intermediate gaps. Therefore, in this final phase the dynamics of the model simply projects the particles into the manifold $\mathcal{M}_t \to \mathcal{M}$. In the generative modeling literature, this phenomenon is also known as *manifold overfitting* as the terms corresponding the internal distribution is negligible (Loaiza-Ganem et al., 2022). In the next section we comment on how these three phases can explain why diffusion models are not affected by this phenomenon, namely why we claim that the manifold is *consolidated* rather than *overfitted*.

## 5.4 The geometric phases and manifold overfitting

The probability density of data defined on a manifold is a spiked object $\delta_{\mathcal{M}}(\boldsymbol{x}) \rho(\boldsymbol{x})$, where the Dirac-delta $\delta_{\mathcal{M}}(\boldsymbol{x})$ determines the manifold and $\rho(\boldsymbol{x})$ determines its internal density. Likelihood-based generative models are defined by a highly parameterized likelihood function $f(\boldsymbol{x}; \boldsymbol{\theta})$, whose parameters are trained by minimizing the loss

$$\mathcal{L}(\boldsymbol{\theta}) = -\mathbb{E}_{\boldsymbol{x} \sim p_0(\boldsymbol{x})}[f(\boldsymbol{x}; \boldsymbol{\theta})] \ , \tag{11}$$

which maximizes the probability of the data given the model. This maximum likelihood loss is minimized if $f(\boldsymbol{x}; \boldsymbol{\theta}) = p_0(\boldsymbol{x})$. A trained likelihood-based model can only fit the true density by having it to diverge to infinity on the manifold. Such a divergence makes it impossible to correctly model the internal density $\rho(\boldsymbol{x})$. More problematically, the optimization problem becomes almost insensitive to the internal density $\rho(\boldsymbol{x})$. This phenomenon is called *manifold overfitting* (Loaiza-Ganem et al., 2022), since the trained model fits the manifold while ignoring its internal density, resulting in poor generation.

Our analysis suggests that the temporal dynamics of generative diffusion models overcome this limitation because, for intermediate values of $t$, the score is still sensitive to the density internal to the manifold, which can be identified through the differences in the tangent singular values. During this manifold coverage phase, the score directs the dispersion of the particles according to these differences, with higher singular values resulting in larger 'opposing force' from the score, which results in smaller displacements of the generated samples along these directions. For $t$ tending to zero, these differences are suppressed due to the divergence of the likelihood, which results in a score function that is orthogonal to the manifold and that is insensitive to $\rho(\boldsymbol{x})$. However, at this stage of generative diffusion the internal dispersion of the particles have already been affected by the previous coverage phase and therefore the manifold overfitting of the score does not negatively affect generation. Instead, the consolidation phase plays the important role of projecting the particles to the support of the data.

## 6 Theoretical analysis of the spectral gaps in linear diffusion models

In this section, we provide our main theoretical results concerning the spectral distribution for random linear subspaces and the relative spectral gaps formulas. We start by reviewing diffusion with data supported on linear manifolds, where the exact score function can be computed.

### 6.1 Linear manifolds

Normally the distribution $p_0(\boldsymbol{x})$ is unknown. It is however interesting to investigate tractable special case where the distribution is a multivariate Gaussian defined as in Eq. 8 where $F \in \mathbb{R}^{d \times m}$ is an arbitrary projection matrix that implicitly define the structure of the latent manifold. and $\boldsymbol{z}^\mu \sim \mathcal{N}(0, I_m)$ the latent space vector. In this setting, the distribution can be explicitly written as $p_0(\mathbf{x}) = \mathbb{E}_{\mathbf{z} \sim \mathcal{N}(0, I_m)} \delta(\mathbf{x} - F\mathbf{z}) = \mathcal{N}(\mathbf{x}; 0, FF^\top)$. Therefore, the density

of the process at a given time $t$ is again Gaussian and can be computed from

$$p_t(\boldsymbol{x}) = \mathbb{E}_{\mathbf{z}} \frac{1}{\sqrt{(2\pi t)^d}} \; e^{-\frac{1}{2t}\|\mathbf{x}-F\mathbf{z}\|^2}. \tag{12}$$

While linear manifolds are very simple when compared with real data, they still exhibit a rich and non-trivial phenomenology that elucidate several universal phenomena of diffusion under the manifold hypothesis. In fact, these linear models capture the structure of tangent spaces of smooth manifolds (see Supp. D).

The score function of the linear model is solvable analytically since we only have to perform Gaussian integrals, from which we obtain a quadratic form in $\mathbf{x}$ that we can rewrite as

$$\log p_t(\mathbf{x}) = \frac{1}{2t}\mathbf{x}^\top J_t\mathbf{x} + const. \tag{13}$$

where the constant does not depend on $\mathbf{x}$ and

$$J_t = \frac{1}{t}F\left[I_m + \frac{1}{t}F^\top F\right]^{-1} F^\top - I_d. \tag{14}$$

The score function is thus derived as $\nabla_{\mathbf{x}} \log p_t(\mathbf{x}) = \frac{1}{t}J_t\mathbf{x}$. It is then useful to analyze the spectrum of the matrix $J_t$, since $J_t$ is proportional to the Jacobian of the score function. In fact, since the gradient of the score is orthogonal to the manifold sufficiently close to it, the number of null eigenvalues of $J_t$ will correspond to the manifold dimension and we should expect to see a drop in the spectrum.

In the following, we provide an outline of our theoretical results on the distribution of spectral gaps in the matrix $J_t$ under random linear manifolds. This choice reflects the fact that the distribution and support of the data are usually not known in advance, and it is therefore important to quantify the statistical variability induced by this uncertainty. We will consider two different distributions for the random projection matrices $F$: an isotropic Gaussian case, and a multiple-variance one. To ensure tractability, we perform the analysis in the limit of large $d$ (visible) and $m$ (latent) dimensions while keeping the ratio $\alpha_m = m/d$ constant.

## 6.2 THE ISOTROPIC CASE

If the elements of the projection matrix $F$ are sampled as $F_{ij} \sim \mathcal{N}(0, \sigma^2/m)$, we are able to derive analytically the full expression of the distribution of the eigenvalues of $J_t$. It is given by a simple transformation of the distribution of the eigenvalues of $F^\top F$, which is known to be the Marchenko-Pastur distribution reported in Supp. A.1.

In Fig. 7 we show the shape of the spectrum at different times. The bulk of the distribution, inherited from the density of the eigenvalues of $F^\top F$, gradually shifts from left to right in the support. By measuring the cumulative function of the spectrum, one can isolate a drop in the effective dimensionality of the manifold, as also plotted in Fig. 7. The step is present at any time in the process and it is implied by the gap between the left bound of the bulk and the spike in $-1$. The width of this gap evolves in time according to

$$\Delta_{\text{fin}}^{\text{GAP}}(t;\sigma) = \frac{\sigma^2(1+\alpha_m^{-1/2})^2}{t + \sigma^2(1+\alpha_m^{-1/2})^2}. \tag{15}$$

If we name $\gamma_+(\sigma)$ the left bound eigenvalue of the bulk in the spectrum of $F^\top F$ (see Fig. 6), one can recover a more general expression for the gap, being

$$\frac{\gamma_+(\sigma)}{t + \gamma_+(\sigma)} = \Delta. \tag{16}$$

Hence we can resolve the gap at a scale $\Delta$ at the time

$$t_{\text{in}} = \gamma_+(\sigma)\left(\frac{1-\Delta}{\Delta}\right). \tag{17}$$

### 6.3 Intermediate gaps and subspaces with different variances

Another relevant case for our study is the one where we consider a manifold having multiple subspaces with different variances. Here we will focus on the instance of two distinct variances. This scenario is reproduced by considering a number $f \cdot m$ of columns of $F$ to have element Gaussian distributed with zero mean and variance $\sigma_1^2/m$, and the remaining $(1-f) \cdot m$ columns with elements from a Gaussian with variance $\sigma_2^2/m$. The spectrum of $J_t$ can be computed also in this case, as explained in Supp. A.2. The density function of the eigenvalues shows a transient behavior of the spectrum in the form of an intermediate drop in the estimated dimensionality of the hidden data manifold. This behavior is reported in Fig. 2. Even though the expression of the spectral density doesn't have an explicit analytical form and has to be computed numerically, one can adopt a special assumption on the behavior of the density of the eigenvalues of $F^\top F$ to estimate the typical times at which the intermediate drop occurs. Generally speaking, the spectrum of $F^\top F$ can be composed of two separated bulks, as observable in Fig. 8. This happens when $\sigma_2^2$ and $\sigma_1^2$ are significantly different. In analogy with the single variance scenario, we name $\gamma_+$ the left bound of the bulk associated with higher eigenvalues, i.e. with the higher variance, and $\gamma_-$ the right bound of the bulk associated with smaller eigenvalues, i.e. smaller variance. Most commonly, $\gamma_- = \gamma_-(\sigma_2)$ and $\gamma_+ = \gamma_+(\sigma_1)$. In this case the gap-forming phenomenology provides for two distinct processes: the emergence of intermediate gaps (i.e. steps in the dimensionality plot) between separated bulks of the spectrum, the opening of a final gap that allows to infer the dimensionality of the full manifold. The width of the intermediate gap between two bulks can be obtained from Eq. (26) as

$$\Delta_{\text{inter}}^{\text{GAP}}(t; \sigma_1, \sigma_2) = \frac{t}{t + \gamma_-(\sigma_2)} - \frac{t}{t + \gamma_+(\sigma_1)}. \tag{18}$$

By imposing $\Delta_{\text{inter}}^{\text{GAP}}(t; \sigma_1, \sigma_2) = \Delta$ one finds the following quadratic form

$$\Delta t^2 + \left[(\Delta - 1)\gamma_- + (\Delta + 1)\gamma_+\right]t + \Delta\gamma_-\gamma_+ = 0. \tag{19}$$

Considering $\Delta \ll 1$ and $\gamma_+ \ll \gamma_-$ the opening time for the intermediate gap can be found by

$$t_{\text{in}}(\Delta) \simeq \Delta^{-1}\gamma_-(\sigma_1), \tag{20}$$

that is a reference time at which the gap becomes visible. On the other hand, by assuming the closure time to be close to zero, it can be obtained as

$$t_{\text{fin}}(\Delta) \simeq \Delta\gamma_+(\sigma_2). \tag{21}$$

Furthermore, the time at which the gap is maximum in width, and so maximally visible, is located in between $t_{\text{in}}$ and $t_{\text{fin}}$. This is the most important time scale for the problem, it is obtained by imposing $\partial\Delta^{\text{GAP}}/\partial t = 0$ and it measures

$$t_{\text{max}} = \sqrt{\gamma_-(\sigma_1)\gamma_+(\sigma_2)}. \tag{22}$$

Indeed, when $\sigma_1^2 \gg \sigma_2^2$ the total spectrum can be approximated by a mixture of two separated Marchenko-Pastur distributions, with variances $\sigma_1^2$ and $\sigma_2^2$, and parameters $\alpha_m$ and $\gamma$ to be rescaled with respect to $f$ and $(1-f)$. This approximation becomes exact under a slight modification of $F$ which does not imply any loss of the quality of the description. Now the relevant quantities for the gap become

$$\Delta_{\text{inter}}^{\text{GAP}}(t; \sigma_1, \sigma_2) = \frac{t\left[f\sigma_1^2(1 - \sqrt{\frac{1}{f\alpha_m}})^2 - (1-f)\sigma_2^2(1 + \sqrt{\frac{1}{(1-f)\alpha_m}})^2\right]}{\left[t + (1-f)\sigma_2^2(1 + \sqrt{\frac{1}{(1-f)\alpha_m}})^2\right]\left[t + f\sigma_1^2(1 - \sqrt{\frac{1}{f\alpha_m}})^2\right]} \tag{23}$$

$$t_{\text{in}}(\Delta) \simeq \Delta^{-1}f\left(1 - \sqrt{\frac{1}{f\alpha_m}}\right)^2\sigma_1^2, \qquad t_{\text{fin}}(\Delta) \simeq \Delta(1-f)\left(1 + \sqrt{\frac{1}{(1-f)\alpha_m}}\right)^2\sigma_2^2, \tag{24}$$

$$t_{\text{max}} = \sqrt{f(1-f)}\left(1 - \sqrt{\frac{1}{f\alpha_m}}\right)\left(1 + \sqrt{\frac{1}{(1-f)\alpha_m}}\right)\sigma_1 \cdot \sigma_2. \tag{25}$$

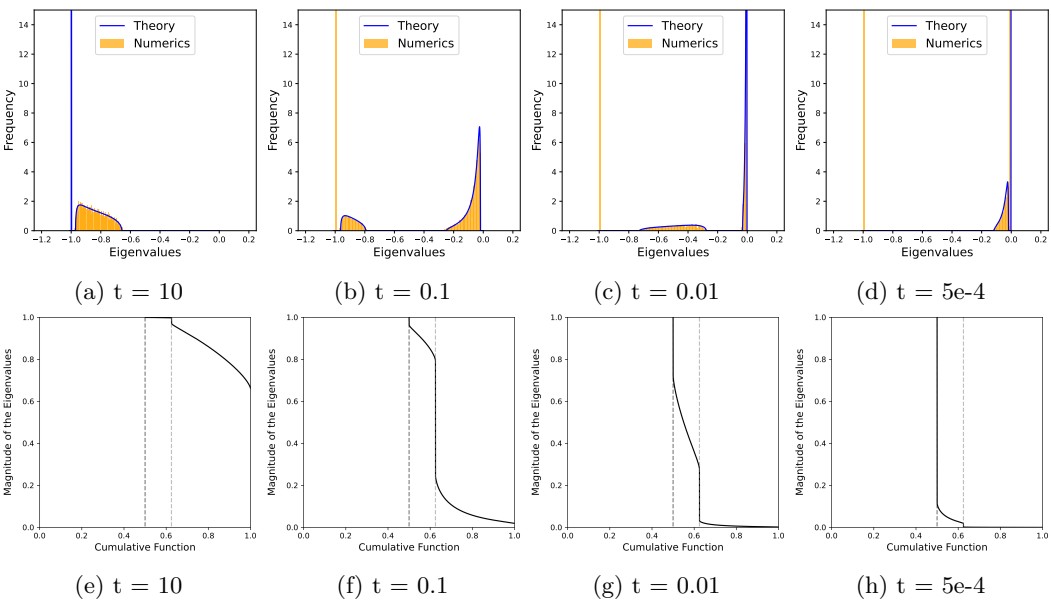

Figure 2: Spectrum of the eigenvalues of $J_t$ and drop in the dimensionality of the data-manifold estimated from theory in the double-variance case, with $\alpha_m = 0.5, \sigma_1^2 = 1, \sigma_2^2 = 0.01$, $f = 0.75$. Numerical data are generated with $d = 100$ and collected over 100 realizations of the $F$ matrix.

This same analysis can be extended to the more general case where the spectral density is known to be formed by different detached bulks, associated with hierarchically smaller variances of the data. The evolution of the intermediate gaps in a double-variance diffusion model is reported in Fig. 2: notice that $t_{\max} = \mathcal{O}(\sigma_2)$ is consistent with Fig. 2b and 2f, where the gap was found to be maximum in width. It is worth noting that subspaces with higher variances are the first ones to be explored by diffusion, and to be learned by the model. This point suggests that the model is sensitive to the parameters of the probability distribution on the manifold, as recently suggested by other works in the literature (see Section 9 for further details).

## 7 Experiments with synthetic linear datasets

We first measure the spectrum of the singular values of the Jacobian of a score function trained through a neural network on a linear manifold data-model generated by two variances $\sigma_1^2$, $\sigma_2^2$ as described in Section 6.3. Results are reported in Fig. 3 (left and central panels). The opening of the gaps is consistent with the theory for the exact score: an intermediate gap associated with the subspace with higher variance first opens; subsequently, the gap relative to the lower variance subspace, which here corresponds to the final gap, opens. We can infer the dimensions of the subspaces by subtracting the location of the dashed vertical lines from $d$. We underline the fact that higher-variance subspaces are learned first by repeating the experiment after swapping the values of the variances. Eventually, the right panel in Fig. 3 reports the same experiments where variances are uniformly generated in the interval $[10^{-2}, 1]$: it is evident that the $d$ intermediate gaps is now a continuous line, as it is expected to be in more realistic natural data-sets. We will now compare the gaps computed analytically with ones obtained from real neural networks trained on the same linear data-model. The results of such comparison are presented in Fig. 4, and they show a good agreement between the ordered distribution of the singular values obtained through empirical methods, and the relative analytical counterpart, computed through the replica method. The opening of the predicted intermediate gaps signal the right dimension of the linear subspaces as verified from the experiments. One can notice from the figure that the analytical profile shows the shape of a sharp step between the zero value along the $x$-axis

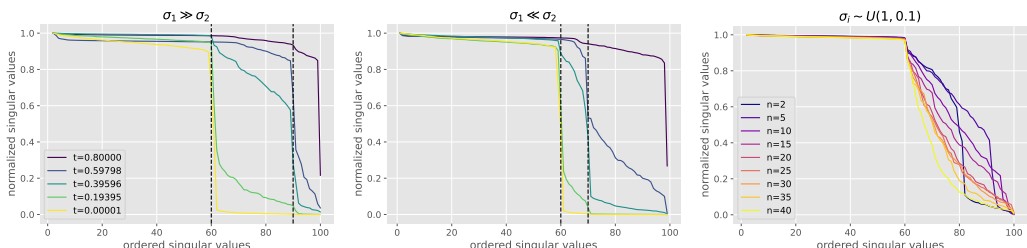

Figure 3: Ordered singular values obtained with the trained score model, for different variances on the subspaces. Data are generated according to the linear-manifold model with $d = 100$ and $m = 40$. Left: $\sigma_1^2 = 1$, $\sigma_2^2 = 0.01$, $f = 0.75$; Center: $\sigma_1^2 = 0.01$, $\sigma_2^2 = 1$, $f = 0.75$; Right: a progressive number $n$ of variances sampled uniformly between $10^{-2}$ and 1 each one assigned to a fraction $f = 1/n$ of matrix columns. The neural network is trained as prescribed in Supp. B and spectra are measured according to Supp. C.

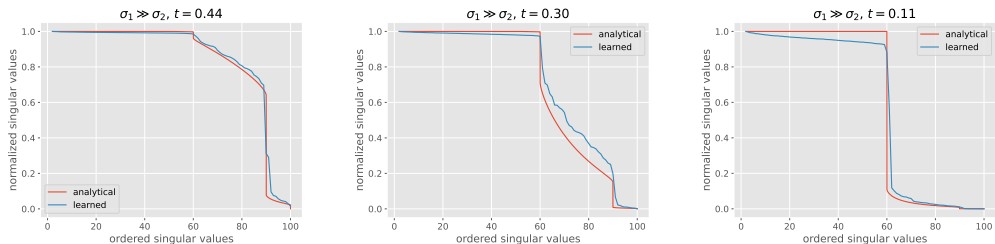

Figure 4: Comparison between spectra obtained with the trained score model and with the numerical analysis, for different variances on the subspaces. Data are generated according to the linear-manifold model with $d = 100$ and $m = 40$, $\sigma_1^2 = 1$, $\sigma_2^2 = 0.01$, $f = 0.75$; from left to right, the spectra are evaluated a time $t \approx 0.45$, $t \approx 0.3$, $t \approx 0.11$. The neural network is trained as prescribed in Supp. B and spectra are measured according to Supp. C.

and the first appearing gap: this shape is related to the Dirac-delta spike that the spectrum of the eigenvalues presents at $-1$ (see Supp. A.2 for details about the spectrum); on the other hand, the numerical profile looks different in the same region, and this behavior is associated with the absence of the spike in the distribution of the singular values, that leaves room to a separated bulk from the other ones. This evident discrepancy between theory and experiment is probably due to the final configuration of the trained neural network and leaves space for further investigations.

## 8 Experiments with natural image datasets

While our theoretical analysis is limited to linear random manifold models, several qualitative aspects of its phenomenology can be observed in networks trained on natural images. Fig. 5 shows the temporal evolution of the spectrum estimated numerically from the Jacobian of models trained on MNIST, Cifar10 and CelebA. Details about the training process are provided in Supp. B and Supp. C. In these experiments we can recognize the three geometric phases of diffusion described above:

**Trivial phase**: at large times (i.e. from $t = 200$ to $100$) the ordered spectrum of the singular values appears flat, suggesting the diffusive motion to be Brownian in the ambient space.

**Manifold coverage phase**: at intermediate times the spectrum shows a clear trace of multiple simultaneous opening gaps. The shape of the curves is, however, different from the controlled scenario showed in Section 6, due to two different reasons: similar latent variances associated to different latent dimensions imply a smoothing of the curve, as displayed by Fig. 3 (right panel); the local eigenspace of the data is complex and hard to model microscopically: for instance, the pixellated appearance of Cifar10 images might explain the scarce emergence

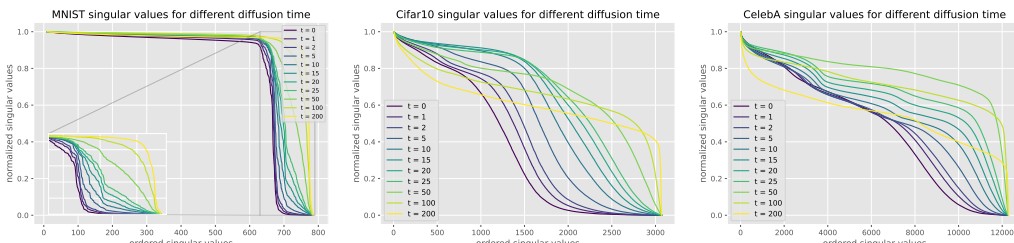

Figure 5: Jacobian spectra of diffusion models trained on MNIST, Cifar10 and CelebA. The neural network is trained as prescribed in Supp. B and spectra are measured according to Supp. C.

of the gaps, while the larger gap structure showed by CelebA might be due to correlations among the latent variances.

**Manifold consolidation phase**: at small times (below $t = 2$) we finally see only the full manifold gap open and progressively sharpening, in the sense that many singular values become exactly zero. The spectra of this phase are the only ones analyzed in Stanczuk et al. (2022) and used to estimate the manifold dimensionality. For instance, we see from Fig. 5 (left panel), that at the end of the reverse process the network trained on MNIST shows a latent dimension $m \sim 100$ that is coherent with Stanczuk et al. (2022).

We conclude that, although our analysis focuses on the local structure of the data manifold (or the stable latent set in diffusion time), it is effectively supported by experiments on real-world complex datasets.

## 9 RELATED WORK & DISCUSSION

The evolution of the fixed-points of the exact score was studied in (Raya and Ambrogioni, 2024) for the analysis of the spontaneous symmetry phenomenon breaking and in (Biroli and Mézard, 2023) for the analysis of memorization and glassy phase transitions. The use of spectral gaps to quantify the dimensionality of the manifold was introduced in (Stanczuk et al., 2022), where the total manifold gap is analyzed. Several recent studies investigated the local linear structure of trained diffusion models. For example, (Kadkhodaie et al., 2024) studied the expansion of the Jacobian of trained models and described it as an optimal geometry-adaptive Harmonic representation. Similarly, (Chen et al., 2024b) characterized the linear expansion of the Jacobian of trained networks and characterized the resulting components in terms of their frequency content. Our work can be seen as a theoretical complement to this more applied line of research, as we provide a comprehensive random-matrix analysis of the phenomenon in tractable models. The dynamic geometry of diffusion manifolds was also investigated in (Chen et al., 2024a) using techniques inspired by research on latent generators such as GANs and autoencoders. Another recent work (Peng et al., 2024) uses parameterized low-rank score functions to show an equivalence between training diffusion model and subspace clustering. In our work, we prove results for the exact score for high-dimensional datasets. Finally, (Sakamoto et al., 2024) studies the dynamic geometry of tubular neighborhoods of the latent manifold and connect their singularities with spontaneous symmetry breaking events. Generative diffusion models exhibit rich geometric structures that have the potential to explain their impressive generative capabilities. Our work introduces the use of random-matrix theory techniques for the analysis of their dynamic local geometry and paves the way for the use of advanced statistical physics techniques, which may in the future unveil more global, topological and non-linear aspects of the dynamic geometry of diffusion generative models.

## Acknowledgements

This publication benefited from European Union - Next Generation EU funds, component M4.C2, investment 1.1. - CUP J53D23001330001.
OnePlanet Research Center acknowledges the support of the Province of Gelderland.
The authors are grateful to Marc Mézard for useful suggestions and discussions.

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

## A   ANALYTICAL DERIVATION OF THE SPECTRUM OF $J_t$

### A.1   SINGLE VARIANCE SCENARIO

We want to compute the spectrum of the matrix in (14). Let us first consider the case in which $F$ is a $d \times m$ matrix with Gaussian entries, and call $\gamma$ the eigenvalues of $FF^\top$.

The function that gives the eigenvalues $r$ of $J_t$ as function of $\gamma$ is

$$r_j = \frac{1}{t}\frac{\gamma_j}{1 + \frac{1}{t}\gamma_j} - 1 = -\frac{t}{t + \gamma_j} \tag{26}$$

Thus, knowing that the distribution of $\gamma$ is Marchenko-Pastur, we can obtain the distribution of $r$

$$\rho_t(r) = -\frac{\alpha_m}{2\pi}\frac{1}{r(1+r)}\sqrt{(r_+ - r)(r - r_-)} + (1 - \alpha_m)\,\delta\,(r+1)\,\theta\left(\alpha_m^{-1} - 1\right) \tag{27}$$

for $r \in [r_-(t), r_+(t)]$, with $r_\pm(t) = -\frac{t}{\left(1 \pm \frac{1}{\sqrt{\alpha_m}}\right)^2 + t}$.

One could ask whether the bulk of $J_t$ separates from $r = -1$ at a discrete time. This separation corresponds to a drop in the histogram of eigenvalues. According to Eq. (27), the bulk is always separated from the spike at finite time $t$, because $(1 + \alpha_m^{-1/2})^2 + t$ for every $t$, and the width of the gap is given by $\Delta^{\mathrm{GAP}}(t) = r_-(t) + 1$

$$\Delta^{\mathrm{GAP}}(t) = \frac{(1 + \alpha_m^{-1/2})^2}{t + (1 + \alpha_m^{-1/2})^2}. \tag{28}$$

With respect to the starting spectrum of $F^\top F$, this condition reads

$$\frac{\gamma_+}{t + \gamma_+} = \Delta \tag{29}$$

so the time when we see the drop at a scale $\Delta$ is $t = \gamma_+^2\,\frac{(1-\Delta)}{\Delta}$.

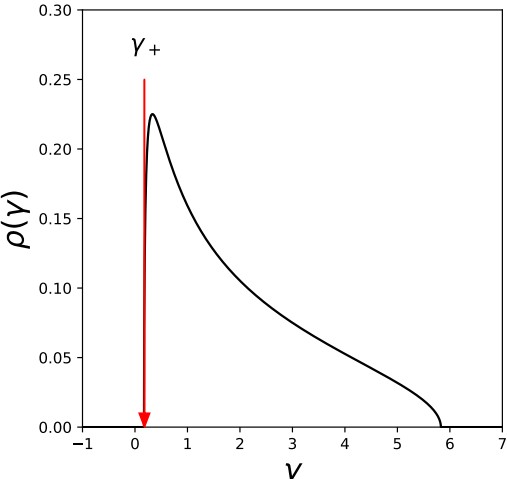

Figure 6: Spectrum of the eigenvalues of $F^\top F$ as obtained from random matrix theory with eigenvalue $\gamma_+$ indicated by red arrow. $\gamma_+$ is provided by the Marchenko-Pastur density function. Control parameters are chosen to be $\alpha_m = 0.5$, $\sigma^2 = 1$.

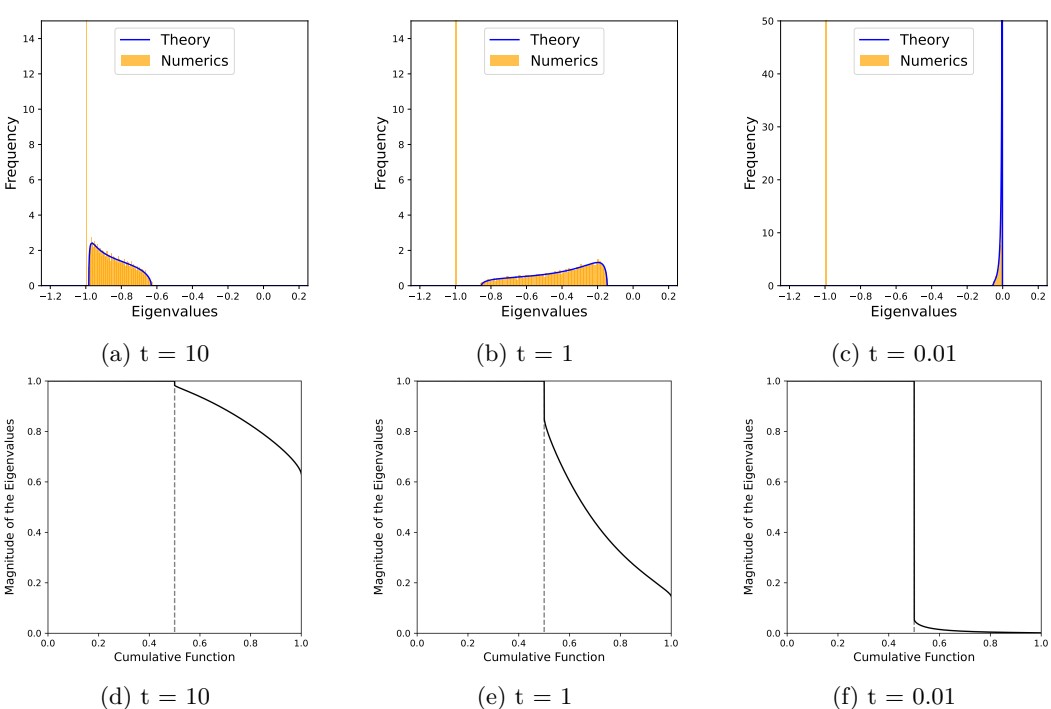

Figure 7: Spectrum of the eigenvalues of $J_t$ and drop in the dimensionality of the data-manifold estimated from theory in the single-variance case, with $\alpha_m = 0.5, \sigma^2 = 1$. Numerical data are generated with $d = 100$ and collected over 100 realizations of the $F$ matrix.

## A.2 Double variance scenario

We want to compute the spectrum of $J_t$ when $F_{i\mu} \sim \mathcal{N}(0, \sigma_1^2)$ for $\mu < fm/2$ and $F_{i\mu} \sim \mathcal{N}(0, \sigma_2^2)$ for $\mu > (1-f)m/2$, with $f \in [0,1]$. We use the replica method to compute the spectrum of $A = \frac{1}{m}FF^\top$, then with a transform we obtain the spectrum of $J_t$. In order to obtain the spectrum we need to compute the expectation of the resolvent of $A$ in the $d \to +\infty$ limit, and to do this we will rely on the replica method

$$\mathbb{E}\left[g_A(z)\right] = -\frac{2}{d}\frac{\partial}{\partial z}\mathbb{E}\left[\log\frac{1}{\sqrt{\det(zI_d - A)}}\right] \tag{30}$$

$$= -\frac{2}{d}\frac{\partial}{\partial z}\lim_{n\to 0}\mathbb{E}\left[\frac{Z^n - 1}{n}\right] \tag{31}$$

with

$$Z^n = \det(zI_d - A)^{-n/2} \tag{32}$$

$$= \int \prod_{a=1}^n \prod_{i=1}^d \frac{d\phi_i^a}{\sqrt{2\pi}}\, e^{-\frac{1}{2}\sum_{a=1}^n\sum_{i,j=1}^d \phi_i^a\left(z\delta_{ij} - \frac{1}{m}\sum_\mu F_{i\mu}F_{j\mu}\right)\phi_j^a} \tag{33}$$

and taking the expectation

$$\mathbb{E}\left[Z^n\right] = \int \prod_{a,i}\frac{d\phi_i^a}{\sqrt{2\pi}}\, e^{-\frac{z}{2}\sum_a\sum_i (\phi_i^a)^2}\mathbb{E}\left[e^{\frac{1}{2m}\sum_a\sum_\mu(\sum_i \phi_i^a F_{i\mu})^2}\right] \tag{34}$$

$$= \int \prod_{a,\mu}\frac{d\eta_\mu^a}{\sqrt{2\pi}}\, e^{-\frac{1}{2}\sum_a\sum_\mu(\eta_\mu^a)^2}\int\prod_{a,i}\frac{d\phi_i^a}{\sqrt{2\pi}}\, e^{-\frac{z}{2}\sum_a\sum_i(\phi_i^a)^2}\prod_\mu\mathbb{E}\left[e^{\frac{1}{\sqrt{m}}\sum_a(\sum_i \phi_i^a F_{i\mu})\eta_\mu^a}\right] \tag{35}$$

where in the last step we have used the independence of the rows of $F$ and applied a Hubbard-Stratonovic transform.

We can separate the product over $\mu$ and integrate over the distribution of $F$

$$\mathbb{E}\left[Z^n\right] = \int\prod_{a,\mu}\frac{d\eta_\mu^a}{\sqrt{2\pi}}e^{-\frac{1}{2}\sum_{a,\mu}(\eta_\mu^a)^2}\int\prod_{a,i}\frac{d\phi_i^a}{\sqrt{2\pi}}e^{-\frac{z}{2}\sum_{a,i}(\phi_i^a)^2} \tag{36}$$

$$\times \prod_{\mu=1}^{fm-1}\mathbb{E}\left[e^{\frac{1}{2\sqrt{m}}\sum_a(\sum_i \phi_i^a F_{i\mu})\eta_\mu^a}\right]\prod_{\mu=fm}^m\mathbb{E}\left[e^{\frac{1}{2\sqrt{m}}\sum_a(\sum_i \phi_i^a F_{i\mu})\eta_\mu^a}\right] \tag{37}$$

$$= \int\prod_{a,\mu}\frac{d\eta_\mu^a}{\sqrt{2\pi}}\, e^{-\frac{1}{2}\sum_{a,\mu}(\eta_\mu^a)^2}\int\prod_{a,i}\frac{d\phi_i^a}{\sqrt{2\pi}}e^{-\frac{z}{2}\sum_{a,i}(\phi_i^a)^2} \tag{38}$$

$$\times e^{\frac{\sigma_1^2}{2m}\sum_i\sum_{\mu<fm}(\sum_a \phi_i^a\eta_\mu^a)^2 + \frac{\sigma_2^2}{2m}\sum_i\sum_{\mu\geq fm}(\sum_a \phi_i^a\eta_\mu^a)^2} \tag{39}$$

$$= \int\prod_{a,\mu}\frac{d\eta_\mu^a}{\sqrt{2\pi}}\, e^{-\frac{1}{2}\sum_{a,\mu}(\eta_\mu^a)^2}\int\prod_{a,i}\frac{d\phi_i^a}{\sqrt{2\pi}}\, e^{-\frac{z}{2}\sum_{a,i}(\phi_i^a)^2}) \tag{40}$$

$$\times e^{\frac{\sigma_1^2}{2m}\sum_{ab}(\sum_i \phi_i^a\phi_i^b)(\sum_{\mu<fm}\eta_\mu^a\eta_\mu^b) + \frac{\sigma_2^2}{2m}\sum_{ab}(\sum_i \phi_i^a\phi_i^b)(\sum_{\mu>fm}\eta_\mu^a\eta_\mu^b)}. \tag{41}$$

Introducing $q_{ab} = \frac{1}{d}\sum_i \phi_i^a\phi_i^b$

$$\mathbb{E}\left[Z^n\right] = \int \prod_{a,b} \frac{dq_{ab}d\hat{q}_{ab}}{2\pi} \int \prod_{ai} \frac{d\phi_i^a}{\sqrt{2\pi}} \, e^{-\sum_{ab} \frac{1}{2}\hat{q}_{ab}\left(dq_{ab}-\sum_i \phi_i^a \phi_i^b\right) - \frac{z}{2}\sum_a \sum_i (\phi_i^a)^2} \tag{42}$$

$$\times \left[\int \prod_{a=1}^{n} \frac{d\eta^a}{\sqrt{2\pi}} \, e^{-\frac{1}{2}\sum_a (\eta^a)^2 + \frac{\sigma_1^2}{2\alpha_m}\sum_{ab} q_{ab}\eta^a \eta^b}\right]^{fm} \tag{43}$$

$$\times \left[\int \prod_{a=1}^{n} \frac{d\eta^a}{\sqrt{2\pi}} \, e^{-\frac{1}{2}\sum_a (\eta^a)^2 + \frac{\sigma_2^2}{2\alpha_m}\sum_{ab} q_{ab}\eta^a \eta^b}\right]^{(1-f)m} \tag{44}$$

$$= \int \prod_{a,b} \frac{dq_{ab}d\hat{q}_{ab}}{2\pi} \, e^{nd\Phi(q,\hat{q})} \tag{45}$$

with

$$\Phi(q,\hat{q}) = -\frac{1}{2n}\sum_{a,b} q_{ab}\hat{q}_{ab} + G_S(\hat{q}) + f\alpha_m G_E(q,\sigma_1) + (1-f)\alpha_m G_E(q,\sigma_2) \tag{46}$$

where

$$G_S(\hat{q}) = \frac{1}{n}\log \int \prod_{a=1}^{n} \frac{d\phi^a}{\sqrt{2\pi}} \, e^{-\frac{z}{2}\sum_a (\phi^a)^2 + \frac{1}{2}\sum_{ab} \hat{q}_{ab}\phi^a \phi^b} \tag{47}$$

$$G_E(q,\sigma) = \frac{1}{n}\log \int \prod_{a=1}^{n} \frac{d\eta^a}{\sqrt{2\pi}} \, e^{-\frac{1}{2}\sum_a (\eta^a)^2 + \frac{\sigma^2}{2\alpha_m}\sum_{ab} q_{ab}\eta^a \eta^b} \tag{48}$$

Using the replica symmetric ansatz $q_{ab} = \delta_{ab}q$, $\hat{q}_{ab} = -\delta_{ab}\hat{q}$

$$G_S(\hat{q}) = -\frac{1}{2}\log(z+\hat{q}) \tag{49}$$

$$G_E(q,\sigma) = -\frac{1}{2}\log(1 - \frac{\sigma^2 q}{\alpha_m}). \tag{50}$$

Putting all together we have

$$\Phi(z) = \frac{1}{2}\hat{q}q - \frac{1}{2}\log(z+\hat{q}) - f\frac{\alpha_m}{2}\log\left(1 - \frac{\sigma_1^2 q}{\alpha_m}\right) - (1-f)\frac{\alpha_m}{2}\log\left(1 - \frac{\sigma_2^2 q}{\alpha_m}\right). \tag{51}$$

The integral can be evaluated by the saddle point method

$$q = \frac{1}{z+\hat{q}} \tag{52}$$

$$\hat{q} = -f\frac{\alpha_m \sigma_1^2}{\alpha_m - \sigma_1^2 q} - (1-f)\frac{\alpha_m \sigma_2^2}{\alpha_m - \sigma_2^2 q}. \tag{53}$$

We can find the Stieltjes transform

$$\mathbb{E}[g_A(z)] = -2\alpha_m \frac{\partial \Phi(z)}{\partial z} \tag{54}$$

$$= \alpha_m q^*(z) \tag{55}$$

where $q^*$ is found by solving the saddle point equation

$$zq^3 + q^2\left(\alpha_m - 1 - \frac{z\alpha_m}{\sigma_1^2} - \frac{z\alpha_m}{\sigma_2^2}\right) +$$
$$+ q\left(\frac{\alpha_m^2}{\sigma_1^2\sigma_2^2}(z - f\sigma_1^2 - (1-f)\sigma_2^2) + \frac{\alpha_m}{\sigma_1^2} + \frac{\alpha_m}{\sigma_2^2}\right) - \frac{\alpha_m^2}{\sigma_1^2\sigma_2^2} = 0. \quad (56)$$

The asymptotic distribution of eigenvalues can be obtained from the Stieltjes transform as

$$\rho(\gamma) = \frac{1}{\pi}\lim_{\epsilon\to 0^+}\text{Im}\left(g_A(\gamma - i\epsilon)\right) \quad (57)$$

$$= \frac{1}{\pi}\lim_{\epsilon\to 0^+}\text{Im}\Phi'(\gamma - i\epsilon) \quad (58)$$

$$= \frac{1}{\pi}\alpha_m\lim_{\epsilon\to 0^+}\text{Im}[q^*] \quad (59)$$

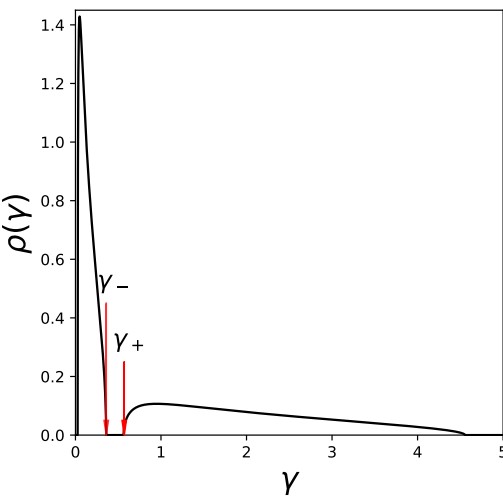

Figure 8: Spectrum of the eigenvalues of $F^\top F$ as obtained from random matrix theory with eigenvalues $\gamma_-$ and $\gamma_+$ indicated by red arrows. Control parameters are chosen to be $\alpha_m = 0.5$, $f = 0.5$, $\sigma_1^2 = 1, \sigma_2^2 = 0.1$.

Once the density of the eigenvalues is computed one can perform the same change of variables described in Supp. A.1 for the case of single variance, and obtain the density $\rho_t(r)$ for the eigenvalues of $J_t$.

Fig. 2 reports the evolution in time of the spectral density, as well as its cumulative function $f = 0.75$. The cumulative function has been used to estimate the formation of the intermediate gaps to compare with the experiments for the estimation of the data-manifold dimension.

## B  Network training and Model Architecture Details

| Dataset | Image Size | Latent Dim. | Channel Mult. | Param. Count | Batch size | Iterations |
|---------|-----------|-------------|---------------|--------------|------------|-----------|
| Cifar10 | 32 | 128 | (1, 2, 2, 2) | 35.7M | 128 | 500,000 |
| MNIST | 28 | 128 | (1, 2, 2) | 24.5M | 128 | 400,000 |
| CelebA | 64 | 64 | (1, 1, 2, 2, 4, 4) | 27.4M | 64 | 800,000 |

Table 1: Table displaying both model and training configurations for each dataset.

For the image datasets, we used the diffusion setting in (Ho et al., 2020). We use the variance scheduler with $\beta_{\min} = 10^{-4}$ and $\beta_{\min} = 2 \times 10^{-2}$, $T = 1000$ time steps, and score model

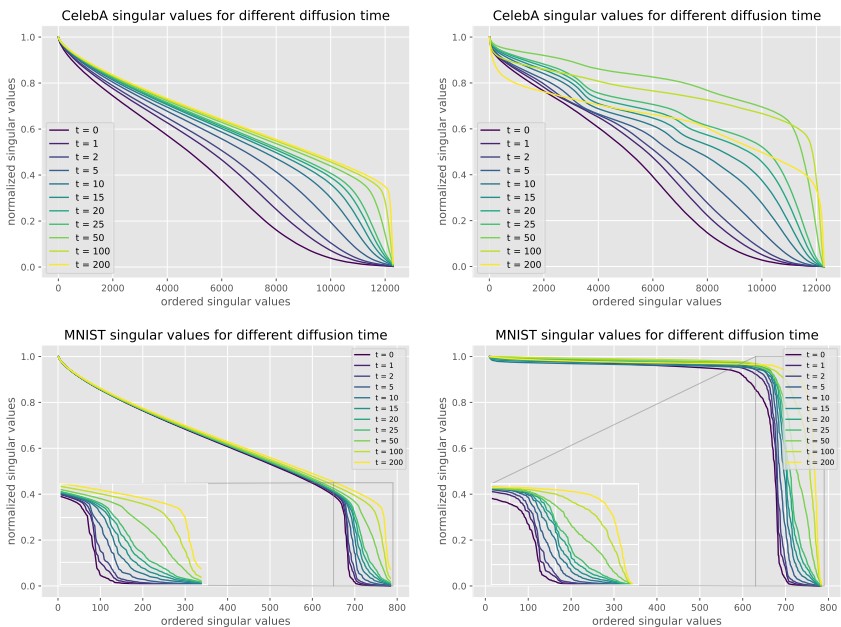

Figure 9: Comparison between the standard SVD method employed by Stanczuk et al. (2022) (left) and the new orthogonalized method for the visualization of the intermediate gaps (right) for a neural network trained on the CelebA and MNIST data-set as described in Section B. Spectra are measured according to Supp. C.

backbone (PixelCNN++ (Salimans et al., 2017)). Furthermore, for each of the datasets, we adjusted the partameters to account for the different complexity (see Table 1). For each data-set, the number of used training data-points amounts to the full set of data available.

For the linear models, we used a Variance Exploding continuous score model trained with 2M steps (batch size 128). The model had a Residual architecture with size 128 hidden channels in each layer, two residual blocks comprised by two linear layers with SiLu. In this case, the number of used training-data amounts to $2 \cdot 10^5$ synthetic examples generated according to Section 6.1.

For all experiments we primarily utilized NVIDIA Tesla V100 GPUs with 32 GB of memory.

## C  EXPERIMENTAL METHODOLOGY: COMPUTING THE SINGULAR VALUES OF THE JACOBIAN OF THE SCORE FUNCTION

For computing the singular values, we used an improved version of the procedure from (Stanczuk et al., 2022) that is reported in Algorithm 1. As a difference from the methodology described in the literature, we perturb the position $\boldsymbol{x}_0$ where we compute the score function $s_\theta$ along strictly orthogonal directions. Such orthogonal perturbations result from a Gaussian sampling of $d$ vectors that are subsequently orthogonalized: these perturbed images are not far from other examples present in the training-set. Choosing orthogonal perturbations instead of random ones significantly reduces the anomalous slope appearing when plotting the ordered singular values that can hide some intermediate gaps (see Fig. 5 in Stanczuk et al. (2022)). Fig. 9 compares the standard SVD method from the literature with our technique by applying it on model trained through from the CelebA and MNIST data-set.

Moreover, for the linear models and MNIST models we used a symmetrized version which we empirically found to be more stable, reported in algorithm 2.

Once the SVs have been obtained through the algorithms, they will be normalized by the highest value in the set and subsequently ordered in a decreasing fashion. In most cases,

the very first singular value is removed from the set, due to a divergent behaviour that is intrinsic to the neural network training.

---

**Algorithm 1** Estimate singular values at $x_0$

---

**Require:** $s_\theta$ - trained diffusion model (score), $t_0$ - sampling time.
 1: Sample $x_0 \sim p_0(x)$ from the data set
 2: $d \leftarrow \dim(x_0)$
 3: $S \leftarrow$ empty matrix
 4: **for** $i = 1, ..., d$ **do**
 5:      Sample $x_{t_0}^{(i)} \sim \mathcal{N}(x_{t_0}|x_0, \sigma_{t_0}^2 I)$ perturbations
 6: **end for**
 7: $(x_{t_0}^{(i)})_{i=1}^d \leftarrow (\tilde{x}_{t_0}^{(i)})_{i=1}^d$ orthogonalized perturbations.
 8: **for** $i = 1, ..., d$ **do**
 9:      Append $s_\theta(\tilde{x}_{t_0}^{(i)}, t_0)$ as a new column to $S$
10: **end for**
11: $(s_i)_{i=1}^d, (v_i)_{i=1}^d, (w_i)_{i=1}^d \leftarrow \text{SVD}(S)$

---

---

**Algorithm 2** Estimate singular values at $x_0$ with central difference

---

**Require:** $s_\theta$ - trained diffusion model (score), $t_0$ - sampling time.
 1: Sample $x_0 \sim p_0(x)$ from the data set
 2: $d \leftarrow \dim(x_0)$
 3: $S \leftarrow$ empty matrix
 4: **for** $i = 1, ..., d$ **do**
 5:      Sample $x_{t_0}^{+(i)} \sim \mathcal{N}(x_{t_0}|x_0, \sigma_{t_0}^2 I)$ right perturbations
 6:      Sample $x_{t_0}^{-(i)} \sim \mathcal{N}(x_{t_0}|x_0, \sigma_{t_0}^2 I)$
 7:      $x_{t_0}^{-(i)} \leftarrow 2x_0 - x_{t_0}^{-(i)}$ left perturbations
 8: **end for**
 9: $(x_{t_0}^{+(i)}, x_{t_0}^{-(i)})_{i=1}^d \leftarrow (\tilde{x}_{t_0}^{+(i)}, \tilde{x}_{t_0}^{-(i)})_{i=1}^d$ orthogonalized perturbations
10: **for** $i = 1, ..., d$ **do**
11:      Append $\frac{s_\theta(\tilde{x}_{t_0}^{+(i)}, t_0) - s_\theta(\tilde{x}_{t_0}^{-(i)}, t_0)}{2}$ as a new column to $S$
12: **end for**
13: $(s_i)_{i=1}^d, (v_i)_{i=1}^d, (w_i)_{i=1}^d \leftarrow \text{SVD}(S)$

---

## D  LINEAR MANIFOLD MODEL HYPOTHESIS

The manifold hypothesis is a fundamental concept in machine learning. It states that $d$-dimensional data lie on a lower $m$-dimensional manifold embedded in the high-dimensional space, often called the *ambient* space (Fefferman et al., 2016). Whether it exists, such manifold is supposed to present a local curvature in the ambient space, i.e. it is not representable as a $d$-dimensional hyperplane. The hidden manifold model proposed by Goldt et al. (2020) is able to reproduce data $\boldsymbol{y}^\mu$ that are embedded in a latent $m$-dimensional space, as

$$\boldsymbol{y}^\mu = \Phi\left(F\boldsymbol{z}^\mu\right), \tag{60}$$

where $\Phi(\boldsymbol{x})$ is a non-linear function of the $d$-dimensional vector $\boldsymbol{x}$, $F \in \mathbb{R}^{d \times m}$ is a rectangular matrix that projects latent vectors $\boldsymbol{z}^\mu$ on the manifold. Nevertheless, in order to apply the replica method from statistical physics and compute the distribution of the eigenvalues of the Jacobian of the score-function, we constrain ourselves to the simpler case of a linear-manifold, i.e. $\Phi(\boldsymbol{x}) = \boldsymbol{x}$ (see Section 6.1). Indeed this might sound as a limitation to the reproducibility of our results to more realistic data-sets, e.g. natural images. However we claim that, for ordinary diffusive diffusion models in the variance exploding setup, the theory developed for the linear model still applies to non-linear manifold instances, due to the following reasons:

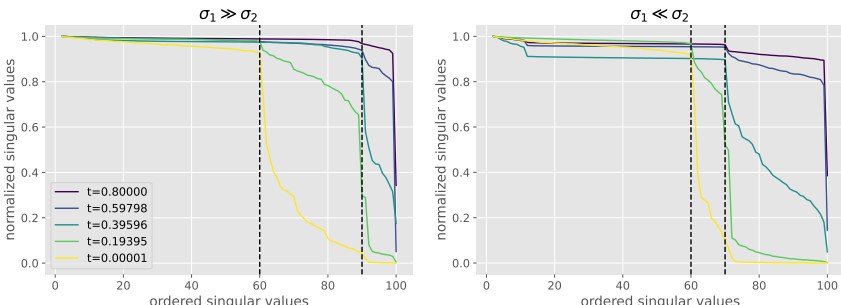

Figure 10: Ordered singular value spectrum estimated experimentally from the analysis of a data-set living on a non-linear manifold built according to Supp. D where parameters are chosen as in Fig. 3. The neural network is trained as prescribed in Supp. B and spectra are measured according to Supp. C.

1. The diffusive trajectories at large $t$, where the *trivial* phase occurs, are sampled by a probability distribution that is smooth, due to the Gaussian kernel implied by the stochastic process in Eq. 4. As a consequence, the stable latent set defined in Eq. 5 will be approximately linear.

2. The diffusive trajectories at small $t$, where the *manifold coverage* and *manifold consolidation* phases occur, explore a region contained into a ball of radius proportional to $\sqrt{t}$, that is supposed to be smaller than then the inverse local curvature of the manifold.

We now compare the distribution of the singular values of the Jacobian obtained from the linear manifold model and the non-linear one. Specifically, we will repeat the plot in Fig. 3 (left and central panels) with a toy data-set generated as $\boldsymbol{y}^\mu = F\tilde{\boldsymbol{z}}^\mu$, with $\tilde{\boldsymbol{z}}^\mu = \boldsymbol{z}^\mu/\|\boldsymbol{z}^\mu\|$. As a consequence, the new data will now live on a $(d-1)$-dimensional ellipsoid inscribed in the original $d$-dimensional hyperplane. The results are reported in Fig. 10 and they show no evident discrepancy between the linear and non-linear case, as predicted by our argument. The only small difference consists of a slight delay in time of the gap phenomenology that is present in the non-linear manifold data which impedes the final gap to fully open at the smallest observable time.

## E  ADDITIONAL EXPERIMENTS

We provide here additional results regarding the disposition of the ordered singular values of the Jacobian of the score function of models trained according to B and C. Specifically, each plot represents the jacobian measured with respect to one single image in the training data-set. We conclude that all images present the same three-phase phenomenology, as a general feature of the data-sets.

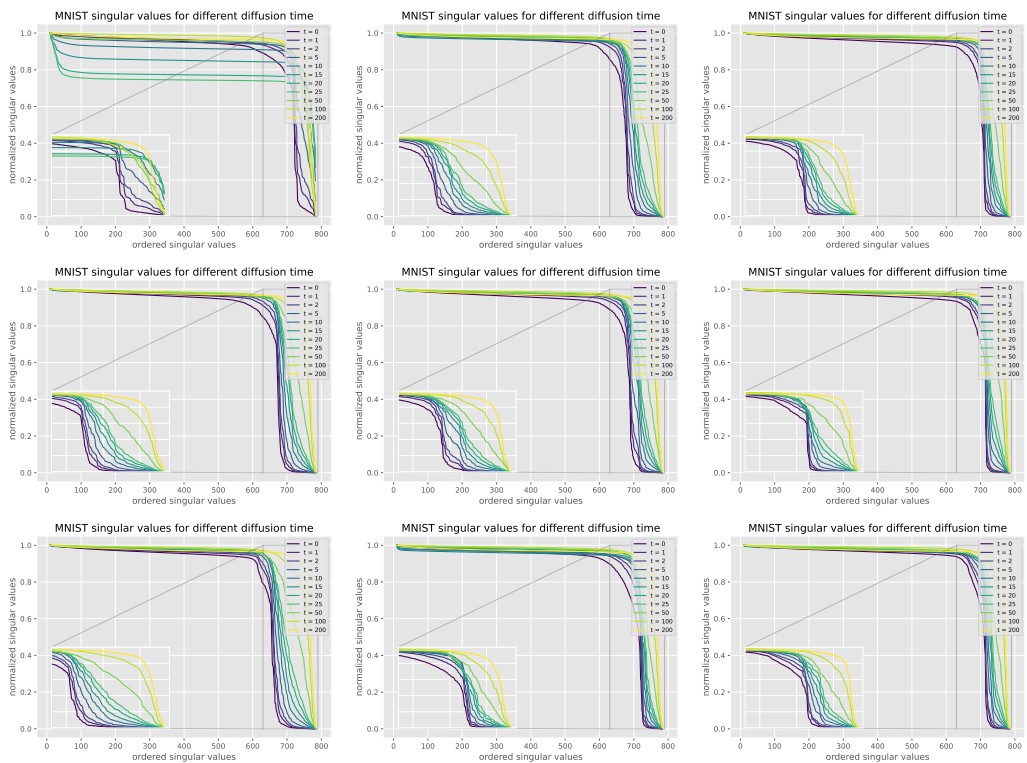

Figure 11: Spectrum of the ordered SVs of the Jacobian for a model trained on MNIST. Each panel is relative to a different data-point in the full set.

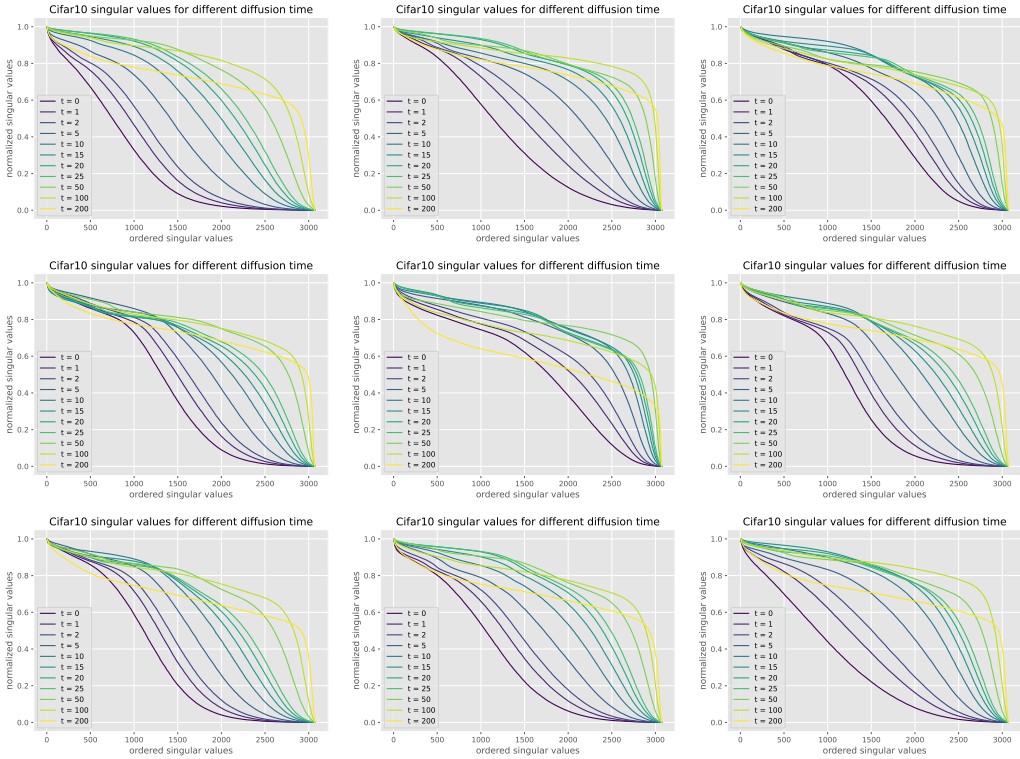

Figure 12: Spectrum of the ordered SVs of the Jacobian for a model trained on Cifar10. Each panel is relative to a different data-point in the full set.

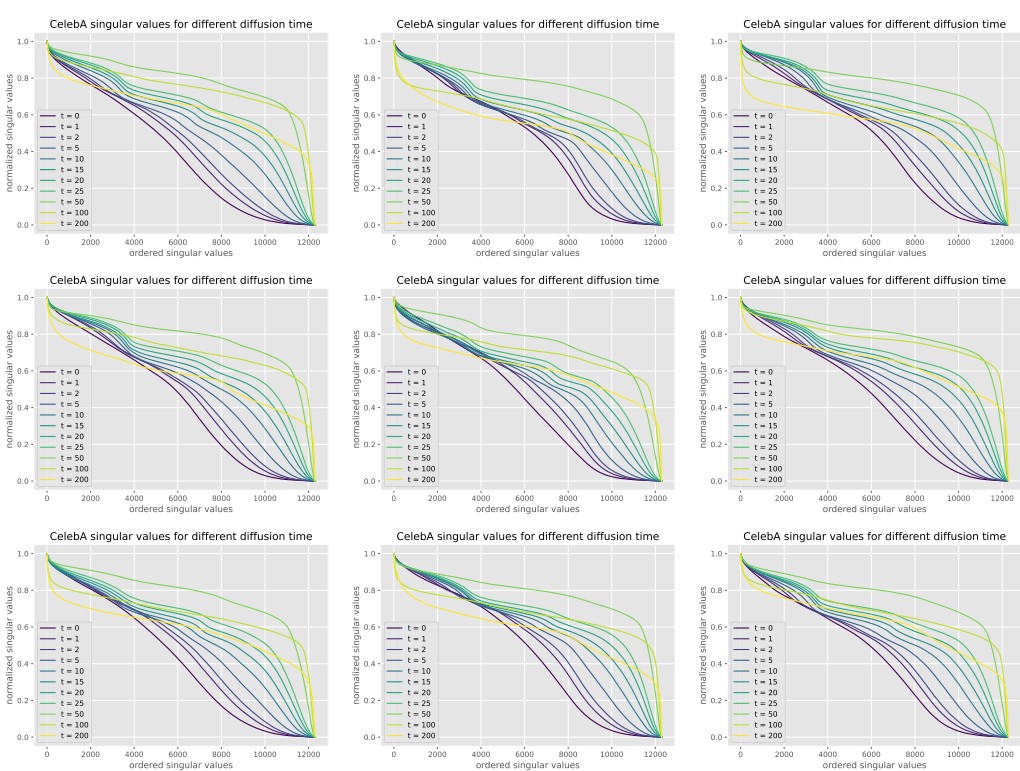

Figure 13: Spectrum of the ordered SVs of the Jacobian for a model trained on CelebA. Each panel is relative to a different data-point in the full set.

