# OpenReview forum: "Manifolds, Random Matrices and Spectral Gaps: The geometric phases of generative diffusion"
_ICLR.cc/2025/Conference — ICLR 2025 Poster_

### Official Review · Reviewer_zg53 · 2024-10-30

**Soundness:** 2
**Presentation:** 1
**Contribution:** 2
**Rating:** 5
**Confidence:** 3

**Summary:**

This paper aims to understand the diffusion models from the geometry of the latent space. The authors did analysis in terms of the spectrum of the score functions' Jacobian, from which the spectral gaps indicates three learning phases of the models. Further, the authors reasoned that the fitting of the distribution and the manifold geometry are disentangled so to explain the indifference of diffusion models to the manifold overfitting.

**Strengths:**

I think this paper is the first work using random matrix to analyze the jacobian in diffusion models, though the idea of analyzing high-dim jacobians using random matrix theory is not new, assuming I read enough literature on understanding diffusion models. Understanding diffusion models, especially from the latent space geometry, is increasingly researched and this paper provides a new way of approaching this analysis.

**Weaknesses:**

1. The paper is delivered in a rather vague way and there are multiple important points which should be clarified in terms of either equations, or citations on other literatures. When carefully reading the paper, I couldn't really parse the ideas at several important locations, though some of these unclear points are only explained later on in a more concrete way in Section 6. I have detailed a few of them in the questions section.
2. The paper strives to better understand the generative diffusion models. While I appreciate the effort using RMT (random matrix theory), the analysis does not really offer deep insights to diffusion models.  In the practical diffusion models, the analysis in Figure 7 merely demonstrated there is a eigenspectrum transition in the Jacobian of the trained model on MINST, which does not have an apparent link to the three phases proposed in the paper. Similar observations have already been shown by _Diffusion Models Encode the Intrinsic Dimension of Data Manifolds_. It would be great if authors could point out the difference or novel insights compared to this work.
3. The paper _Dynamical Regimes of Diffusion Models_ showed the three phases of the generative process in diffusion models using phase transition analysis where these key transitions are also analyzed via the distributional solution of the forward dynamics. These transitions are determined by not only the eigenvalues of the covariance matrix of the data, but also the entropy of the noised data distribution. It really interests me to see the difference between the analysis in this work and the one in _Dynamical Regimes of Diffusion Models_. So it's necessary to point out the new results from this paper compared to the previous analyses.
4. I try to locate in the paper where these different phases explains why diffusion models are prone to manifold overfitting. Unfortunately, I could not find it. It's mentioned in section 2 that likelihood based models lead to increasing model density during training (091-092), thus, resulting in the manifold overfitting. Could authors explain how the three phases of diffusion models help them to be unaffected by the manifold overfitting from this perspective? For example, add a section or paragraph that explicitly connects their three-phase model to the issue of manifold overfitting, explaining step-by-step how each phase contributes to avoiding this problem

**Questions:**

1. (044-045) These decomposition phenomena cannot be directly explained in terms of critical phase transitions as they are fundamentally linear processes.
   - This reasoning is not clear to me.
   - I assume authros wanted to explain the separation of subspaces in diffusion models using the phase transition idea, much like how B. et al. 2024a explained the separation of data classes.
   - what's the main difference of this work from B. et al. 2024a?
2. 091-095 meantioned that the divergence of data distribution implies the model density becomes larger during training. Could authros elaborate on this? Moreover, when claiming diffusion models having three phases are unaffected by manifold overfitting, could authors provide the analysis accordingly in comparison to the likelihood based models?
3. 096-097: it's pointed out in this work, diffusion models are not affected by the manifold overfitting. Could authors provide some evidence on this, e.g., literatures?
4.  140-146: it's not straightforward to see how perturbations aligned to the tangent space of the manifold correcpond to small eigenvalues, while the orthogonal ones correspond to high eigenvalues. Could authors provide more details on this?
5. It's hard to parse Fig 1. Is this at a fixed time t?  I checked the cited paper Stanczuk et al. 2022, but it's still unclear to me. Why the x-axis is latent dimensionality? I do not directly see the link that perturbations more aligned to the tangent space of the manifold are located at the regim of smaller latent dimentionality, not the orthogonal ones have larger dimensionality. How do you find the one-to-one correspondence between the eigenvalues and the latent dimensionality? In Stanczuk et al. 2022, the authors pointed out that the vanishing singular values correspond to the tangent space, while the large ones correspond to the normal space. But it seems there is some nontrivial steps needed from there to here.
7. 172: The phase separation do not correspond to singularities as there are cross-over events, not genuine phase transitions.
   1. what are singularities in this context?
   2. what are cross-over events?
8. It seems $\sigma_k$ is not defined. I assume it is the variance of the data distribution over manifolds with different dimensions. When would $\sigma_{k+1}^2\ll \sigma_k^2$ happen?
9.  What is asymptotic closure? Do you mean the end time of the phase II can be infinite and one considers an approximate of when this closure happens?
10. Please also define $r$ in line 481.

Other suggestions:
- I think the template of the paper is different from the provided one.
- Please correct and unify the bibliography.

---

> ### Author Response · Authors · 2024-11-22
> **Authors Response Part I**
>
> >The paper is delivered in a rather vague way and there are multiple important points which should be clarified in terms of either equations, or citations on other literatures. When carefully reading the paper, I couldn't really parse the ideas at several important locations, though some of these unclear points are only explained later on in a more concrete way in Section 6. I have detailed a few of them in the questions section.
>
> We thank the Reviewer for the useful comments and questions listed below because they have greatly improved our manuscript.
>
> >The paper strives to better understand the generative diffusion models. While I appreciate the effort using RMT (random matrix theory), the analysis does not really offer deep insights to diffusion models. In the practical diffusion models, the analysis in Figure 5 merely demonstrated there is a eigenspectrum transition in the Jacobian of the trained model on MINST, which does not have an apparent link to the three phases proposed in the paper. Similar observations have already been shown by Diffusion Models Encode the Intrinsic Dimension of Data Manifolds. It would be great if authors could point out the difference or novel insights compared to this work.
>
> Stanczuck et al. (2023), describes the phenomenology of diffusion models for small values of $t$, in a regime that we referred to as the manifold-consolidation phase. On the other hand, our analysis investigates how the geometry evolves as a function of time. We show that, under a controlled scenario, at intermediate times it is possible to identify the emergence and disappearance of intermediate gaps that encode the relative variances of the (local) sub-spaces that form the manifold. In accordance with the theory, we see that in more complex scenarios like on MNIST the intermediate phase exhibits a smooth transition instead of a sharp gap, since the different subspaces are not separated by large gaps in their variances but they instead form a gradient. The real bridge between the linear toy data-model and real-world data-models is eventually contained in Figure 3: increasing the number and density of latent variances smoothes the intermediate gaps, as observed for real data (for which we cannot build a closed model).
>
> The new spectral estimation method is described in Supp. C. Figure 5 now better relates to the theory: MNIST, CIfar10 and CelebA models.
> In all these datasets, the ‘trivial phase’ is visible in the large values of $t$, where the spectra do not show any clear manifold structure. The ‘manifold coverage phase’ is visible in the intermediate values of $t$, where there is a clear gradient-like sub-gap structure at intermediate times. Finally, the score at the earliest time point, as investigated in Stanczuck et al. (2023), does not present these features due to manifold overfitting, where the divergence of the orthogonal component of the score suppresses the differences among the sub-spaces.

---

> ### Author Response · Authors · 2024-11-22
> **Authors Response Part II**
>
> >The paper Dynamical Regimes of Diffusion Models showed the three phases of the generative process in diffusion models using phase transition analysis where these key transitions are also analyzed via the distributional solution of the forward dynamics. These transitions are determined by not only the eigenvalues of the covariance matrix of the data but also the entropy of the noised data distribution. It really interests me to see the difference between the analysis in this work and the one in Dynamical Regimes of Diffusion Models. So it's necessary to point out the new results from this paper compared to the previous analyses.
>
> The paper “Dynamical Regimes of Diffusion Models” by Biroli et al. (2024) presents some crucial differences, in both their analysis and scientific question. These points of divergence with respect to our work will be now listed:
> - Our paper studies the geometric evolution of the score vector field in the ambient space of data with respect to a manifold, while this is not the object of study of Biroli et al., which does not assume the manifold hypothesis. While we perform our analysis by evaluating the singular values/eigenvalues of the Jacobian of the score function, these quantities are never considered in the latter paper.
> - Our paper does not provide a critical theory (i.e. a theory about phase transitions) for diffusion models. In fact, the emergence of our phases is not signaled by the breaking of a symmetry. We define these phases as cross-over events, i.e. geometric behaviors of the score function that become more evident at specific times in the backward process. On the other hand, Biroli et al. isolate two critical events in the backward process: speciation, signaled by a symmetry breaking in the configuration space; and collapse, signaled by a condensation transition in an effective random energy model.
> - The cross-over events of our interest are qualitatively different from the dynamic events evaluated by Biroli et al. While our trivial phase coincides with their Regime I, speciation is different from the manifold coverage: the former represents a random choice of the model when choosing a class (among multiple ones) to collapse on, and the latter describes the sampling on a subspace of the latent manifold without caring about the division of the data in different classes (e.g. the class might be unique). Finally, collapse cannot take place in our framework, since we consider the exact score function of the model, rather than the empirical one. The empirical score tends to the exact one when $N\gg d$.
> To conclude, both papers tackle the dynamics of diffusion models yet under two very different perspectives. We thank the Referee for giving us the chance to clarify such differences.
>
> >I try to locate in the paper where these different phases explains why diffusion models are prone to manifold overfitting. Unfortunately, I could not find it. It's mentioned in section 2 that likelihood based models lead to increasing model density during training (091-092), thus, resulting in the manifold overfitting. Could authors explain how the three phases of diffusion models help them to be unaffected by the manifold overfitting from this perspective? For example, add a section or paragraph that explicitly connects their three-phase model to the issue of manifold overfitting, explaining step-by-step how each phase contributes to avoiding this problem
>
> We thank the Reviewer for the feedback. We included a new Subsection (5.4) explaining why diffusion models are not affected by the negative consequences of manifold overfitting due to the separation between the coverage and consolidation phases.

---

> ### Author Response · Authors · 2024-11-22
> **Authors Response Part III**
>
> >(044-045) These decomposition phenomena cannot be directly explained in terms of critical phase transitions as they are fundamentally linear processes.
> This reasoning is not clear to me.
> I assume authros wanted to explain the separation of subspaces in diffusion models using the phase transition idea, much like how B. et al. 2024a explained the separation of data classes.
> what's the main difference of this work from B. et al. 2024a?
>
> We now answer to the Reviewer doubts in order:
> - Let us explain the cited line more explicitly. The dynamical events that we have studied do not satisfy the standard (Landau’s) definition of phase transitions. Such definition implies the breaking of some symmetry, represented by a non-zero value for a so-called order parameter of the model (e.g. the magnetization in ferromagnetic systems, the overlap in glasses and associative memories, the vapor density in ideal gases etc.). This symmetry breaking occurs at a certain value of the so-called control parameter of the model (e.g. temperature for ferromagnetic systems, glasses and gases, the density of patterns in associative memories etc.). Yet our dynamic events appear as emergent behaviors similarly to phase-transitions. We call these “apparent” phase transitions cross-over events. We also state that these phenomena are “linear processes” because they are not separated by any (abrupt or continuous) discontinuity of the order parameter when varying the control parameter (that is time, in our case). Our exact theory concludes that gaps are always opened across time, but such opening becomes measurable only at specific times, i.e. when the gaps are wider than a scale $\Delta$ defined in Eq. (16). This specific criterion tries to replace the presence of an order parameter. Similar cross-over events are present in many physics domains, such as condensed matter, QCD and biophysics (e.g. this topic is treated in D.J. Amit, “Field Theory, the Renormalization Group and Physical Phenomena” (1993)).
> - As stated in a previous answer to the Reviewer’s comments, our approach is not critical, in the statistical mechanics sense, but essentially geometric. Statistical mechanics is exclusively used in our work for computing the Steltjies transform of spectral densities. Hence it is very different from the one employed by Biroli et al. (2024).
> - The main differences between our work and Biroli et al. (2024) have been resumed in the answer to the highlighted weaknesses of the papers, slightly above in this document.
>
> >091-095 meantioned that the divergence of data distribution implies the model density becomes larger during training. Could authros elaborate on this? Moreover, when claiming diffusion models having three phases are unaffected by manifold overfitting, could authors provide the analysis accordingly in comparison to the likelihood based models?
>
> The probability density of data defined on a manifold is a "delta function like" object $\delta(x - f(x)) \rho(x)$, where $f$ determines the manifold and rho its internal density. By training a likelihood-based model $p(x)$, we can only fit the true density by having it to diverge to infinity on the manifold. Such a divergence makes it impossible to correctly model the internal density $\rho(x)$. Our analysis suggests that the temporal dynamics of generative diffusion models overcome this limitation because, for intermediate values of $t$, the score is sensitive to the density internal to the manifold, which can be identified through the differences in the tangent singular values. During this ‘manifold coverage phase’, the score directs the dispersion of the particles according to these differences, with higher singular values resulting in a larger ‘opposing force’ from the score, which results in smaller displacements of the generated samples along these directions. For t tending to zero, these differences are suppressed due to the divergence of the likelihood, which results in a score function that is orthogonal to the manifold and that is insensitive to $\rho(x)$. However, at this stage of generative diffusion, the ‘internal’ dispersion of the particles has already been affected by the previous coverage phase and therefore the manifold overfitting does not negatively affect generation.
>
> >096-097: it's pointed out in this work, diffusion models are not affected by the manifold overfitting. Could authors provide some evidence on this, e.g., literatures?
>
> Apart from the evidence provided by our recent paper, this idea is mentioned in “Denoising deep generative models” Loaiza-Ganem et al. (2022), “Score-based generative models detect manifolds” Pidstrigach (2022) and “Convergence of denoising diffusion models under the manifold hypothesis” De Bortoli et al. (2022).

---

> ### Author Response · Authors · 2024-11-22
> **Authors Response Part IV**
>
> >140-146: it's not straightforward to see how perturbations aligned to the tangent space of the manifold correcpond to small eigenvalues, while the orthogonal ones correspond to high eigenvalues. Could authors provide more details on this
>
> The score is by definition the gradient of the density of the (noised) data distribution. If the data is supported on a manifold, the probability density will decrease sharply if we move from a point in the data manifold along an orthogonal direction, since the probability of the data is by definition zero outside of its support. On the other hand, infinitesimal tangent perturbations result in points that are themselves inside of the manifold and that therefore have higher probability density. This implies that the gradient, and consequently its linearization given by the singular values, is smaller in the tangent directions.
>
> >It's hard to parse Fig 1. Is this at a fixed time t? I checked the cited paper Stanczuk et al. 2022, but it's still unclear to me. Why the x-axis is latent dimensionality? I do not directly see the link that perturbations more aligned to the tangent space of the manifold are located at the regime of smaller latent dimensionality, not the orthogonal ones that have larger dimensionality. How do you find the one-to-one correspondence between the eigenvalues and the latent dimensionality? In Stanczuk et al. 2022, the authors pointed out that the vanishing singular values correspond to the tangent space, while the large ones correspond to the normal space. But it seems there is some nontrivial steps needed from there to here.
>
> We do confirm that Figure 1 shows the ordered eigenspectrum of the Jacobian at a fixed time and position in the ambient space, as an illustrative sketch. The x-axis is named “latent dimensionality” because the non-zero singular values/eigenvalues will span a range on the axis that corresponds to the orthogonal dimensions with respect to the manifold. On the other hand, the null singular values/eigenvalues will span the complementary range, relative to the latent dimensions internal to the manifold. Please refer to our answer to the previous Question.
>
> >172: The phase separation do not correspond to singularities as there are cross-over events, not genuine phase transitions.
> what are singularities in this context?
> what are cross-over events?
>
> We are now going to answer to the Reviewer’s question by order:
> - Singularities are, in the context of Landau’s theory of phase transition, discontinuities in the derivative (of any order) of a free energy function parametrized with respect to an order parameter of the model. This approach is applied, for instance, by Biroli et al. (2024) to explain the speciation and collapse transitions in the backward process for data models with multiple classes.
> - Cross-over events are events that share qualitative similarities with rigorous phase transition which cannot be detected through a singularity in the derivative of a free energy function.
>
> >It seems $\sigma_{k}$ is not defined. I assume it is the variance of the data distribution over manifolds with different dimensions. When would $\sigma_{k+1}^2 \ll \sigma_k^2 $ happen?
>
> $\sigma^2_{k}$ is the variance of the $k$-th linear subspace that we consider as constituent of our linear manifold; namely, it is the variance of a subgroup of columns of the matrix $F$, that are Gaussianly distributed with zero mean. The condition $\sigma_{k+1}^2 \ll \sigma_k^2$ does not happen in the case of many variances that generate $F$, as in real-world scenario. This is the reason why we can't see evident opened subgaps in real cases, but rather smooth curves, as showed in the experiment reported in Figure 5, panel c and by the theory (see Eq. 55). We thank the Reviewer for pointing this out, and we proceed to clarify it in the main text.
>
> >What is asymptotic closure? Do you mean the end time of the phase II can be infinite and one considers an approximate of when this closure happens?
>
> Consistently with the fact that our dynamic phases are signaled by cross-over events instead of proper phase transitions, the intermediate gaps are fully closed only at $t = 0$, at the same time when the final manifold-consolidation gap is fully opened. This is supported by Equation (18) in the main text.
>
> >Please also define r in line 481.
>
> We thank the Reviewer for underlining the missing definition of $r$. We have proceeded with the removal of the symbol $r$ from the main text. This variable is now defined and used only in the Supplementary.

---

> > ### Comment · Reviewer_zg53 · 2024-11-25
> >
> > Dear authors, I appreciate your effort in addressing my comments.
> > While I recognize the novelty of the work, I believe it requires more substantial work to be considered for acceptance due to the following main concerns:
> > 1. the presentation remains poor. In a 10-page paper, there are 9 sections in the paper. While this does not pose as a problem, the lack of connections between sections makes it read as a first draft or a piece of notes by putting observations and initial analyses together. Important definitions like manifold overfitting should be well-laid out in the main text and the main text should read as a standalone paper. You need the manifold projection $F$ already in section 5, while the definition does not appear until section 6. Section 8 on the real-world experiments should be extensively discussed in terms of the previous analyses, which are largely missing. You might want to finish the sentence in line 440.
> > 2. a large discrepancy between the proposed three phases and the actual observations from real-world experiments --- As also pointed out by other three reviewers, the three phases remain intuitive and they are not well-supported by the practical experiments. This is largely due to the limitation of the analysis to linear models. The primary focus on experiments should be about the real-world ones, instead of the synthetic one.
> >
> > Therefore, despite I increased the rating, I do not recommend the paper for acceptance at this stage.

---

> ### Author Response · Authors · 2024-11-26
> **Response to Official comment by Reviewer zg53**
>
> > the presentation remains poor. In a 10-page paper, there are 9 sections in the paper. While this does not pose as a problem, the lack of connections between sections makes it read as a first draft or a piece of notes by putting observations and initial analyses together. Important definitions like manifold overfitting should be well-laid out in the main text and the main text should read as a standalone paper. You need the manifold projection already in section 5, while the definition does not appear until section 6. Section 8 on the real-world experiments should be extensively discussed in terms of the previous analyses, which are largely missing. You might want to finish the sentence in line 440.
>
> Thanks to the Reviewer's comments we have indeed worked a lot to improve our manuscript. Specifically, we have moved the definition of our theoretical linear model before the three phases description, so now the matrix $F$ is correctly defined before it is used. Furthermore, we wrote a full subsection on the manifold overfitting phenomenon in diffusion models compared to likelihood based models, where we explain how our description relates to this phenomenon. This dissertation is now fully included in the main text, instead of being relegated to the Supplementaries. Lastly, we have abundantly expanded Section 8, that comments on how the phases emerge from real data spectra accordingly to the present discussion. We in fact defend our thesis, and provide evidence of the fact that our theoretical prediction of the geometric phases can be found in real data experiments. We also fixed some other typos as the one spotted by the Reviewer at line 440.
>
> > a large discrepancy between the proposed three phases and the actual observations from real-world experiments --- As also pointed out by other three reviewers, the three phases remain intuitive and they are not well-supported by the practical experiments. This is largely due to the limitation of the analysis to linear models. The primary focus on experiments should be about the real-world ones, instead of the synthetic one.
>
> Although our analysis focuses on the local structure of the data manifold and the three geometric phases that we propose are, in more general cases, a conjecture, it is effectively supported by experiments on real-world complex datasets. To support this point, we have expanded Section 8 in the newly uploaded version of the manuscript by adding detailed comments about how one can recover the three phases from the real data spectra.
>
> We sincerely hope that the discussion provided in Section 8 proves insightful to the Reviewer, irrespective of the final judgment on our work. Additionally, we are deeply grateful to the Reviewer for significantly enhancing the quality of our article.

---

> > ### Comment · Reviewer_zg53 · 2024-12-02
> >
> > I thank the authors for their revision and clarifications. I will keep these in mind in the later potential discussions.

---

### Official Review · Reviewer_Cqtz · 2024-11-01

**Soundness:** 3
**Presentation:** 2
**Contribution:** 2
**Rating:** 6
**Confidence:** 3

**Summary:**

This paper investigates the spectral properties of generative diffusion models within the framework of the manifold hypothesis. By analyzing the spectrum of the Jacobian of the score function, the authors identify and interpret three distinct phases in the generative process: trivial phase, manifold coverage phase, and manifold consolidation phase. These phases explain the internal distribution alignment and resilience against manifold overfitting, which is often seen in likelihood-based models. Through a mix of theoretical analysis using linear manifolds and random matrix theory, alongside high-dimensional experiments on image datasets, the paper demonstrates how these phases manifest and validate the theory.

**Strengths:**

* Clear exposition, good fit in the existing literature
* The analysis of the Jacobian spectrum and its phase-wise evolution provides a meaningful interpretation of the manifold structure and how diffusion models capture it
* Using linear manifolds and random matrices enables tractable and insightful theoretical analysis supporting the findings.
* Experiments on complex datasets, such as MNIST and CIFAR-10, align with theoretical predictions, showing the three observable phases.

**Weaknesses:**

* Introducing the mathematics for manifolds earlier could clarify the manifold overfitting problem and improve the logical flow.

See questions

**Questions:**

* In section "The linear random-manifold score in large dimensions: A random matrix analysis", is the approach aimed to study the behaviour of diffusion models when encountering random datasets?
* In Equation (5), $M_t$ appears to represent the manifold where noised data reside. Is the idea that particles are found in shells of radius $\sqrt{t}$ around the latent manifold consistent with this? If $M_t$ is not merely a union of spheres centered on data points (I don't think I believe it is the case), what precisely is its mathematical function? I understand you want to consider a specific space on which the noised data lies, in order to use the intepretations deriving from (6), but I don't think introducing $M_t$ is a good mathematical strategy

---

> ### Author Response · Authors · 2024-11-22
> **Authors Response**
>
> >Introducing the mathematics for manifolds earlier could clarify the manifold overfitting problem and improve the logical flow.
>
> We thank the Reviewer for the precious advice that will improve the presentation of our manuscript. Specifically, we added an introduction about data-manifolds from Section 2.
>
> >In section "The linear random-manifold score in large dimensions: A random matrix analysis", is the approach aimed to study the behaviour of diffusion models when encountering random datasets?
>
> In Section 6.2 we introduce the results we obtained for the linear manifold model presented in Section 6.1. This model does not aim to study random datasets. Instead, it is a theoretical model introduced by Goldt et al. (2020) to generate data from an hidden manifold, that we have restricted to the linear manifold case.
>
> >In Equation (5), $M_t$ appears to represent the manifold where noised data reside. Is the idea that particles are found in shells of radius $\sqrt{t}$ around the latent manifold consistent with this? If $M_t$ is not merely a union of spheres centered on data points (I don't think I believe it is the case), what precisely is its mathematical function? I understand you want to consider a specific space on which the noised data lies, in order to use the intepretations deriving from (6), but I don't think introducing $M_t$ is a good
>
> In order to study the latent geometry of diffusion for times larger than $t$, it is important to be able to define the correct geometric quantities in question. At time $t \to 0$, the manifold can be clearly identified as the set of fixed points of the score, at least asymptotically. This manifold structure is generally also visible for finite times. However, at time $t$ the latent manifold is not identical to the manifold where the data is supported, since the high-frequency variations are suppressed (‘smoothed out’) by the forward kernel. This implies that the latent manifold at time $t$ is different from the manifold at time zero, and studying its dynamic geometry is one of the central aims of the current work. We extended and reformulated the explanation of our definitions in Section 4 to make them more clear and to explain their rationale.

---

> > ### Comment · Reviewer_Cqtz · 2024-12-02
> >
> > I acknowledge the authors rebuttal. I maintain my score.

---

### Official Review · Reviewer_jgnp · 2024-11-02

**Soundness:** 3
**Presentation:** 3
**Contribution:** 3
**Rating:** 6
**Confidence:** 4

**Summary:**

This paper provides a theoretical perspective on generative diffusion models under the manifold hypothesis, focusing on the spectrum of the score function's Jacobian. The authors propose that spectral gaps indicate the presence and dimensionality of distinct sub-manifolds. They derive analytical forms of the spectrum under the assumption that the manifold is a linear subspace, revealing three phases in the generative process: a trivial phase, a manifold coverage phase, and a consolidation phase. THey argue that this division explains why generative models avoid the manifold overfitting issues seen in likelihood-based models.

**Strengths:**

1. The perspective on generative diffusion models, focusing on the spectrum of the score function's Jacobian, appears novel.

2. The three-phase framework in generative diffusion models provides insight into model behavior across different generative stages.

3. The approach is supported by mathematical derivations under a linear assumption and is further validated with illustrative examples on toy datasets like MNIST.

**Weaknesses:**

1. The linear assumption is overly restrictive and unlikely to hold in practical scenarios, limiting the generalizability of the theoretical results.
2. The real data results on CIFAR-10 and CelebA (Figure 7) are difficult to interpret and lack sufficient clarity to be fully convincing.
3. In Section 5, the conjecture that "their phenomenology captures the main features of subspace separation in the tangent space of curved manifolds" lacks a clear intuition or justification, making it unconvincing based on the current presentation.

A minor issue:
There are some citation formatting issues. For example. the third reference in the first paragraph of the intro, should be Song et al, instead S et al. See also the B. et all in the same paragraph, and so on.

**Questions:**

Following the above weakness, I have the following questions.

1. Can the authors either show some results for nonlinear manifold? If not, at least certain discussion is expected.
2. Can the authors further discuss the results on CIFAR-10 and CelebA? These results are not sufficient to support the claim in the paper, nor the conjecture.
3. This is not really a question. At a first glance I thought this is a theoretical paper, but after reading it carefully I feel it's more like a summary and conjecture based on empirical observation, as the theoretical part is only restricted to linear models.

---

> ### Author Response · Authors · 2024-11-22
> **Authors Response Part I**
>
> >The linear assumption is overly restrictive and unlikely to hold in practical scenarios, limiting the generalizability of the theoretical results.
>
> As the Reviewer noticed, the linear manifold case we study is indeed limited. However, it has two significant advantages that we would like to emphasize here. Firstly, it is analytically solvable, allowing us to compare the behavior of a neural network-approximated score with the true score in a scenario that, while simple, still provides some data structure. Additionally, it serves as a good approximation for diffusion models at both small and large times. We have expanded on this in Supp. D, but in essence, this approximation holds because, at small $t$, trajectories stay close to the manifold, making it approximately linear, and at large $t$, they are drawn from a probability distribution smoothed by a Gaussian kernel, which again results in an approximately linear structure.
> Furthermore, while our analytical solution is restricted to the linear case, we aim to experimentally explore more complex scenarios, including natural images. Recognizing the importance of connecting the linear case to more general non-linear settings and real-world data, we have taken steps to clarify this connection by:
> - Adding plots of the Jacobian spectrum of a neural network score as the number of (linear) subspaces with varying variances gradually increases (Fig. 3c), illustrating how the gap phenomenology becomes progressively smoothed.
> - Including additional plots (Fig. 10) similar to those in Fig. 3, but incorporating a small non-linear effect.
> Thanks to the Reviewer’s comments, we have made a concerted effort to elucidate the relationship between our linear analytical results and the empirical findings on real datasets.
>
> >The real data results on CIFAR-10 and CelebA (Figure 7) are difficult to interpret and lack sufficient clarity to be fully convincing.
>
> We have indeed improved the presentation of these results, and we hope the Reviewer will find them more convincing. Notice that the major change we made is in the method that we are adopting to compute the spectrum of the Jacobian, as we now restrict to orthogonal perturbations, as described in Supp. C. This results in a less noisy spectrum, where the intermediate gaps associated with the manifold coverage phase can be now envisaged, despite the smoothing effect displayed in Figure 3, panel c. Results on some real data sets, such as Cifar10 do not show sharp spectral gaps, probably due to the blurry appearance of the images: pixelated data points might result in a less sharp manifold structure. However, in spite of the lack of a sharp gap, the manifold consolidation phase is still detectable by the fact that a large set of singular values is approximately zero for small values of $t$. These are the eigenvalues corresponding to the tangent components, whose relative differences in variance are highly suppressed at small values of t while it is visible at intermediate times (i.e., in the manifold coverage phase). Interestingly, CelebA shows a peculiar manifold-coverage phase, where intermediate gaps are wide and evident: we conjecture that this behavior might be due to a particular hierarchical structure of the latent variances in the manifold, and it deserves future investigations. We further discuss the results on Cifar10 and CelebA in the newly added Supp. E.
>
> > In Section 5, the conjecture that "their phenomenology captures the main features of subspace separation in the tangent space of curved manifolds" lacks a clear intuition or justification, making it unconvincing based on the current presentation.
>
> We thank the Reviewer for giving us the opportunity to discuss this aspect more in detail. We have then added a Supplementary Section (Supp. D) where we provide a theoretical argument that supports the validity of the linear manifold model even in non-linear data models. This argument is supported by the literature (see Statczuk et al. (2023)) and by numerical experiments that we also report in the same section. The results show a remarkable alignment between the linear theory and the non-linear results even for intermediate values of $t$.

---

> ### Author Response · Authors · 2024-11-22
> **Authors Response Part II**
>
> >Can the authors either show some results for nonlinear manifold? If not, at least certain discussion is expected.
>
> We thank the Reviewer for the interest in non-linear results. Unfortunately, we do not know how to solve analytically the non-linear manifold case. However, to answer the Reviewer’s question, have added an experiment with synthetic data where the latent variables are generated on an ellipsoid in the ambient space (Fig. 10). Adding such a non-linearity does not change much the phenomenology of the linear case, consistently with the discussion we have provided in Supp. D. We have tried to comment on the importance of the linear case by answering the "Weaknesses" section.
>
> >Can the authors further discuss the results on CIFAR-10 and CelebA? These results are not sufficient to support the claim in the paper, nor the conjecture.
>
> We have tried to comment more on these results answering the "Weaknesses" section.
>
> >This is not really a question. At a first glance I thought this is a theoretical paper, but after reading it carefully I feel it's more like a summary and conjecture based on empirical observation, as the theoretical part is only restricted to linear models.
>
> This paper provides both a thorough theoretical analysis of the linear case and a set of experiments indicating that the linear phenomenology is preserved in many non-linear scenarios. As we showed in our work, a proper spectral analysis of the linear case already requires complex random-matrix techniques to be fully carried out and we think it is an important first step in the development of a more general geometric theory of generative diffusion.

---

> > ### Comment · Reviewer_jgnp · 2024-11-24
> >
> > Thank the authors to the detailed responses. The newly added discussion on CIFAR-10 and CelebA looks good. However, my concern on the linear assumption remains. I totally understand that extensions to general (nonlinear) manifold is very challenging. Actually myself works on ML/DL theory, that's why I gave a positive score (6) even noticing this very strong, and often unrealistic assumption.
> >
> > As a result, I decide to keep my positive score for now, and will discuss with other reviewers and the AC.

---

### Official Review · Reviewer_fVvn · 2024-11-03

**Soundness:** 3
**Presentation:** 2
**Contribution:** 2
**Rating:** 5
**Confidence:** 4

**Summary:**

The paper provides a random-matrix theory derived result computing the exact eigenspectra of the score of a diffusion modeling a distribution supported on a linear submanifold of a sample space. This distribution is a Gaussian supported on a hyperplane in Euclidean space. A three phase process on the eigenvalues is observed in the diffusion placed on the modeled density. This theoretical computation matches the observation of a number of previous works studying the eigenspectra of the score function of diffusion models.

**Strengths:**

- The results are of some interest and validate observational results of the change in distribution of the eigen spectra of the score function of diffusion models through time.

**Weaknesses:**

- The setting studied is a rather limited setting, and given the linear nature of the submanifold the density is supported on, and the Gaussian nature of the density.
- The experimental section of the paper is very light and leaves many questions unexplored.

**Questions:**

- The experimental detail of section 7 is very light and lacks much detail needed for recreation. For example the number of samples from the distribution from training is missing.
- Section 7 leaves a poignant question unexplored - how does the number of samples from the data distribution used to train the score function affect the empirical results in estimating the dimension of the data manifold. The distribution of eigenvalues obtained in the results appear to not mach perfectly the theoretical results. Is this due to modeling issues, the amount of data used, or some other issues? Exploring a range of dataset sizes may shed some light on why the image experiments fail to show the definitive eigenspectra cliff needed to detect the intrinsic data dimensionality. Perhaps these datasets simply don't contain enough examples to learn the score well.
- If the conclusion of section 8 is that for real datasets the proposed method of looking at the distribution of eignevalues of the score function fails to detect the intrinsic dimensionality of the data, can the authors comment on how this menthod might be of practical utility in training or understanding diffusion models?

---

> ### Author Response · Authors · 2024-11-22
> **Authors Response Part I**
>
> >The setting studied is a rather limited setting, and given the linear nature of the submanifold the density is supported on, and the Gaussian nature of the density.
>
> As pointed out by the Reviewer, the linear manifold case we study is of course limited, nonetheless, it has two very important strengths, that we would like to underline here:
>
> - Firstly, it is solvable analytically, so that we can compare the behavior of a neural network approximated score with the true score in a case which, although simple, provides some data structure.
> - It is also a good approximation for diffusion models at both small and large times. We have added Supp. D to further elaborate on this point, but in simple words, this happens because at small t trajectories are close to the manifold, thus it is approximately linear, while at large t they are sampled from a probability distribution which is smoothed by a Gaussian kernel, so, again, approximately linear. Both our manifold-coverage and manifold-consolidation phases occur at small times.
>
> Moreover, while it is true that we analytically solve only the linear case, we aim to study experimentally more complex cases, such as natural images. Since we understand that the connection to more general non-linear cases and natural images is fundamental, we decided to clarify it by adding:
> - Plots of the spectrum of the Jacobian of a neural network score in the case where the number of (linear) subspaces with different variances is gradually increasing (Fig. 3c) to show how the gap phenomenology we have described becomes progressively smoothed
> - The same plots of Fig. 3 but adding a small non-linear effect (Fig. 10)
> Thanks to the Reviewer’s comments, we have made an effort to elucidate the link between the linear analytical results and the empirical findings on real datasets.
>
> >The experimental section of the paper is very light and leaves many questions unexplored.
>
> The experimental section of the paper aims to validate our theory for the linear manifold case in the following settings:
> - When data come from a linear manifold model, and the score function is approximated by a neural network and not the exact one.
> - When data are real.
> We believe that the experiments cover the fundamental situations to show that there is good agreement between theory and neural network behavior and that the phenomenology extends to the more complex and interesting case of real datasets. In the typical physics approach, we focus on a toy model that is able to give good insights into real-world applications. As we were explaining in the previous comment, we have added a few more results, that we hope can provide a clearer picture.
> Regarding unexplored questions, we will try to address them in the “Questions” section.

---

> ### Author Response · Authors · 2024-11-22
> **Authors Response Part II**
>
> >The experimental detail of section 7 is very light and lacks much detail needed for recreation. For example the number of samples from the distribution from training is missing.
>
> We thank the Reviewer for poining this out, we have added this information in the main text. Indeed, regarding Section 7, we previously failed to mention that we generated 20k datapoints from the linear manifold model for training in each of the presented situations. For what concerns the experiments on real datasets described in Section 8, we have added more details in Supp. B. In particular, for training we used the complete datasets.
>
> >Section 7 leaves a poignant question unexplored - how does the number of samples from the data distribution used to train the score function affect the empirical results in estimating the dimension of the data manifold. The distribution of eigenvalues obtained in the results appear to not mach perfectly the theoretical results. Is this due to modeling issues, the amount of data used, or some other issues? Exploring a range of dataset sizes may shed some light on why the image experiments fail to show the definitive eigenspectra cliff needed to detect the intrinsic data dimensionality. Perhaps these datasets simply don't contain enough examples to learn the score well.
>
> In (old) Fig. 6 the discrepancy is not imputable to the lack of expressivity of the network, but rather to the way the Jacobian is computed. Indeed, for the Jacobian of the score learned with a network we used the method described by Stanczuk et al. (2023), mainly to align with the literature on how manifold dimension can be computed in diffusion models. To address the Reviewer’s point more clearly, we have added a new version of these plots, where the Jacobian of the neural network approximated score is obtained perturbing along orthogonal, instead of random, directions. This method is further described in Supp. C. Now the agreement is much more evident, and we hope these new plots will help clarify this aspect. We have also applied this orthogonalized method to the real image datasets.
> With respect to the influence of the number of samples, the recent paper "Losing dimensions: Geometric memorization in generative diffusion" by Achilli et al. (2024)  shows that a sufficiently high number of training points is needed to correctly infer the dimensionality of the stable latent set. We made sure to match this condition in our experiments as well.
>
> >If the conclusion of section 8 is that for real datasets the proposed method of looking at the distribution of eignevalues of the score function fails to detect the intrinsic dimensionality of the data, can the authors comment on how this menthod might be of practical utility in training or understanding diffusion models?
>
> As stated in the manuscript, the paper offers an explanation of the high performance of generative diffusion models for manifold-supported data. As a result of our analysis, we argue that manifold overfitting is avoided by a sort of ‘division of labor’, where different aspects of the generated data (i.e. the geometry of the manifold and its internal density) are learned at different time points during diffusion. On the other hand, other likelihood-based models such as VAEs cannot solve this problem as they learn a single density at a fixed noise level. While we do not currently have direct suggestions on how to leverage these insights to improve training, we are confident that increasing our mathematical understanding of generative diffusion processes will provide algorithmic improvement in the future.

---

### Official Review · Reviewer_Q3Mt · 2024-11-04

**Soundness:** 3
**Presentation:** 3
**Contribution:** 3
**Rating:** 6
**Confidence:** 3

**Summary:**

The paper studies the generative process of diffusion models, meaning how samples of the target distribution are generated starting from random initial values, by looking at spectral gaps in the Jacobian matrix of the score function.

The manifold hypothesis is adopted and, in order to conduct a mathematic analysis, the authors consider the simplified case of linear manifolds and Gaussian target measures. In this setting, the temporal evolution of the spectra of the Jacobian is computed either analytically or through a Replica computation, and the resulting spectral gaps are used to estimate the dimensionality of the data-manifold.

The main result of the paper is that, during the generative process, there is a series of spectral gaps opening, each corresponding to a different local subspace (this phase is denoted Manifold Coverage). Then, the more the process gets close to t=0, the more these gaps shrink, except from the principal one, which is the only one sharpening and remaining asymptotically.

The authors verify this theoretical prediction on synthetic datasets, by learning the score with a Neural Network and looking at the empirical distribution of the respective Jacobian.

Finally, some experiments on image datasets are presented, to look at whether the phenomenology described for linear manifolds is valid, in some way, also for more realistic settings.

**Strengths:**

- The paper is fairly-well written.
- The authors adopt an interesting new perspective on the study of generative diffusion processes.

**Weaknesses:**

- Section 7: The justification of the discrepancy in Fig. 6 between the analytical (replica) and the empirical results is weak, in my opinion. If the Neural Network is expressive enough to learn the exact score, shouldn't we expect the two curves to overlap? If one does the same experiments for the isotropic case for which an analytic solution is available, does one get the same discrepancy?
- Section 8: The results for Cifar10 and CelebA do not show the appearance of any spectral gap. I didn't understand the justification at the end of Section 8. Also for MNIST, while a spectral gap seems to appear at small values of $t$, it seems to indicate a dimensionality of the manifold of around 700. How is this compatible to the fact that the dataset considers only 10 classes?
- [minor]: In the Introduction, the Jacobian is mentioned, but it should be said it is the Jacobian of the score.

The paper presents many typos, here is a list of the ones I found (there could be many more):
- Line 40: models should be modes
- Line 66: idealize should be idealized
- Line 113: is is
- Line 126: Eq. (2) should be Eq. (1)
- Line 229: the the
- Line 357: spearated
- Line 501: temputal
- Line 510: suble

I suggest a careful reading to correct any additional typo present.

**Questions:**

- I find the definition of the "support score", and later of the "stable latent manifolds" in Eq. (5), very confusing. Why are the manifolds defined in this way? What is the role of this definition for the rest of the paper? It seems to me that from Eq. (6) onward, this support score is not mentioned anymore and we come back to the normal score.
- Line 131: "radius $\sqrt{t}$" should be "radius proportional to $\sqrt{t}$"?
- I don't know how to reconcile the phenomenology described in Section 5 with the results presented in the later sections. Indeed, from the plots shown in Fig. 3 and 4, it seems that at all times we are in the so-called "Manifold Coverage" phase, with the intermediate gaps open. Does this mean that the "trivial phase" is only at $t=\infty$ and the "consolidation phase" is only at $t=0$? In general, I think the point of separation between these three phases is not well defined.

---

> ### Author Response · Authors · 2024-11-22
> **Authors Response Part I**
>
> > Section 7: The justification of the discrepancy in Fig. 6 between the analytical (replica) and the empirical results is weak, in my opinion. If the Neural Network is expressive enough to learn the exact score, shouldn't we expect the two curves to overlap? If one does the same experiments for the isotropic case for which an analytic solution is available, does one get the same discrepancy?
>
> The discrepancy between the analytical and the empirical results depends on two factors: 1) finite size effects (the analytical predictions are valid in the so-called thermodynamic limit, i.e. $d \to \infty$); 2) the numerical method employed to measure the Jacobian in the experiments. Whereas it is harder to find a solution to the first point, we addressed point 2) and proposed a finer analysis. The empirical calculation of the Jacobian is inspired by the analysis of Statczuk et al. (2023): a trained neural network fits the score function on a set of points living in the data-space; such points are sampled from random perturbations around the origin; then scores are collected as columns of a matrix $J$, and a SVD is performed to obtain the “gap plots”, e.g. (old) Figure 6. Inspired by the Referee’s comment, we have improved this method by orthogonalizing the perturbations employed to compute the score function. This new approach avoids spurious correlations among the columns of $J$ (that now are only d in number) that are responsible for the slope in the gap plots, i.e. an anomalous mass in the spectrum of the eigenvalues. This new technique, explained in Supp. C, aligns the shape of such empirical spectra to the shape of the analytical ones (as reported in Figure 4, upper line) and allows for a proper comparison with the exact results. The same analysis would extend to the isotropic case, a case-study suggested by the Referee.
> >Section 8: The results for Cifar10 and CelebA do not show the appearance of any spectral gap. I didn't understand the justification at the end of Section 8. Also for MNIST, while a spectral gap seems to appear at small values of , it seems to indicate a dimensionality of the manifold of around 700. How is this compatible to the fact that the dataset considers only 10 classes?
>
> The real data-sets evaluated in our analysis must have a more complex manifold structure with respect to the toy-model tackled by the theory. We expect to see clear intermediate gaps only when there is a large separation in variance between different sub-spaces. In more realistic scenarios, a large number of small gaps opens at different time scales, resulting in a smooth curve. A fundamental connection between the toy model and the real world lies in Figure 3, panel c (that we have improved for a better comprehension of the results). This figure shows the evolution of the “gap plot” when the complexity of the model increases, i.e. the manifold projector matrix $F$ is generated according to an increasing number of variances, not necessarily well separated from each other. This case aims at reproducing a more realistic scenario. The result is a progressive smoothing of the single intermediate gap due to the overlapping of multiple, smaller gaps that are opened at the same time in the backward process. This phenomenon is predicted by the theory when Eq. (56) is extended to contain multiple different variances and it is visible in Figure 5, panel a. Cifar10 and CelebA data-set, however, are more difficult to interpret. This must be due to the particular structure of the manifold (and the latent variances) in real data: the intermediate gaps in CelebA show a possible hierarchical structure of the variances, while the absence of the gap structure in Cifar10 might be caused by the blurry, pixelated appearance of the images. Finally, we must point out that the extractable dimensions for MNIST is not 700, as stated by the Referee, but rather 800-700 = 100. This number is compatible with previous estimates in the literature. We must underline, however, that the data manifold dimension does not have a univocal definition in the literature. This quantity, in fact, can be estimated globally (e.g. by PCA, Probabilistic-PCA techniques), or locally in the ambient space (e.g. by Diffusion Models as in Statczuk et al. (2023), LLE etc.): the dispersion of the estimations in the literature, as resumed by Table 2 in Statczuk et al. (2023), is very high. Nevertheless, local techniques find a local dimensionality that is around 100, according to our experiments.
>
> We have also fixed the typos that the Reviewer noticed.

---

> ### Author Response · Authors · 2024-11-22
> **Authors Response Part II**
>
> > I find the definition of the "support score", and later of the "stable latent manifolds" in Eq. (5), very confusing. Why are the manifolds defined in this way? What is the role of this definition for the rest of the paper? It seems to me that from Eq. (6) onward, this support score is not mentioned anymore and we come back to the normal score.
>
> Indeed, our analysis uses the score of the data, not the support score. Unfortunately, it is not possible to precisely define $M_t$ using the score of the data since, when the distribution is not uniform, the score both determines the manifold structure and the probability distribution internal to the manifold. Since the main aim of the paper is to study the temporal evolution of the latent geometry during generative diffusion, we believe it is important to provide a definition of the latent manifold (or set) at intermediate times. Our definition is consistent with the fixed-point analysis used in (Raya & Ambrogioni, 2024). This is used implicitely, since the Jacobian analysis considers perturbations around stable points where the score vanishes (or almost vanishes), so that the linearization and its singular value spectrum determines the local dynamics.
>
> >Line 131: "radius $\sqrt{t}$" should be "radius proportional to $\sqrt{t}$"?
>
> From the self-averaging property of the stochastic process that moves that trajectory of the backward process, the position of each “particle” must concentrate on a shell of radius $\sqrt{t}$ around each data-point. As a consequence, the distance of the particle from each point of the stable set will be proportional to $\sqrt{t}$ when d is finite, converging to $\sqrt{t}$ in the large $d$ limit. We thank the Reviewer for underlining this inaccuracy, that has been fixed in the main text.
>
> >I don't know how to reconcile the phenomenology described in Section 5 with the results presented in the later sections. Indeed, from the plots shown in Fig. 3 and 4, it seems that at all times we are in the so-called "Manifold Coverage" phase, with the intermediate gaps open. Does this mean that the "trivial phase" is only at $t=\infty$ and the "consolidation phase" is only at $t=0$? In general, I think the point of separation between these three phases is not well defined.
>
> The Reviewer is indeed right when objecting that, strictly speaking, “the "trivial phase" is only at $t=\infty$ and the "consolidation phase" is only at $t=0$”. As claimed by our paper, such phases must not be intended in the Landau’s meaning studied by Statistical Mechanics, but rather as cross-overs, i.e. significantly different behaviors of the system displayed at different values of the control parameter, that for us is the time t. While in theory all gaps are always present at finite times, in practice they are not detectable in all phases as their magnitude is too small, as it is clear from our experiments and theoretical calculations. For this purpose we introduce the scale variable $\Delta$ in Eq. (16): this quantity relates to the minimum gap that would be measurable in a more realistic scenario. Our criterion to separate the three different phases then follows: when none of the intermediate gaps are wider than $\Delta$, we are in the trivial case; when at least one intermediate gap is wider than $\Delta$ we are in our manifold-coverage phase; when the final gap is wider than $\Delta$, than we enter the last, manifold-consolidation phase.

---

> > ### Comment · Reviewer_Q3Mt · 2024-11-26
> >
> > I thank the authors for addressing my questions and doubts in detail.
> > I feel that the revision, especially of section 7, makes the paper stronger and I am raising my score accordingly.
> > I would have appreciated it if the authors had highlighted the changes made in the new version to make it easier to see where to look in the revised version.

---

### Author Response · Authors · 2024-11-22
**Response to All Reviewers**

We thank all the reviewers for their thoughtful and helpful feedback. We are pleased that:

**Reviewer Q3Mt** thinks that our paper is fairly-well written, and most importantly that we adopt an interesting new perspective on the study of generative diffusion processes.

**Reviewer fVvn** finds our results of some interest since they validate observational results.

**Reviewer jgnp** finds novel our perspective on generative diffusion models, focusing on the spectrum of the score function's Jacobian that allows to isolate a three-phase phenomenology in generative diffusion models providing insights into the model behavior across different generative stages.

**Reviewer Cqtz** appreciated our theoretical analysis of the Jacobian spectrum and its phase-wise evolution and its alignment with experiments on complex real-world data-sets.

**Reviewer zg53** underlines that this paper is the first work using random matrix to analyze the jacobian in diffusion models

More importantly, we thank the reviewers for carefully identifying some significant weaknesses and avenues for improvement, which pushed us to perform new experiments and undergo a significant rewriting of the paper to deliver our points in a clearer fashion. We are currently working on the revised manuscript, and will upload it as soon as possible.

In the meantime, we wanted to address the weaknesses and questions raised by each reviewer in individual responses below.
Please do not hesitate to let us know if you have additional comments or questions, which will allow us to achieve the best possible version of our paper.

---

### Meta-Review · Area_Chair_WQru · 2024-12-19

**Metareview:**

Summary:
This paper presents a novel theoretical analysis of generative diffusion models through the lens of differential geometry and random matrix theory. By examining the spectral gaps in the Jacobian of the score function, the authors identify three distinct phases of generative diffusion: a trivial phase, a manifold coverage phase, and a consolidation phase. This work provides valuable insights into the generative process, particularly in understanding how diffusion models avoid manifold overfitting—a common issue in likelihood-based models. The theoretical analysis is supported by experiments on synthetic and real-world datasets.

Strengths:

Novel Perspective: The use of random matrix theory to analyze the Jacobian in generative diffusion models is novel, contributing a fresh theoretical angle to the field.

Insightful Contributions: The identification of the three geometric phases offers an explanation for why diffusion models do not suffer from manifold overfitting, which has been a critical issue in generative modeling.

Rigorous Theoretical Foundation: The paper develops analytical tools for the linear manifold case, providing a clear and elegant connection between theory and empirical observations.

Improved Clarity and Experiments: Following reviewer feedback, the authors significantly revised the paper, clarifying key points, addressing methodological concerns, and expanding the discussion of real-world experiments.

Weaknesses:

Simplified Assumptions: The linear manifold assumption limits the generality of the theoretical results. While the authors provide justification and some experiments to extend the findings to nonlinear manifolds, these remain conjectural.

Experimental Limitations: Some reviewers found the experiments on real-world datasets to be insufficiently linked to the theoretical claims, with weaker evidence of the three-phase structure in CIFAR-10 and CelebA datasets.

Presentation Concerns: The initial presentation of the paper was noted to be fragmented and difficult to follow. While the revised version addresses many of these concerns, there is room for further improvement in connecting sections and streamlining definitions.

Discussion:

The reviewers largely agree on the novelty and value of the theoretical contributions, particularly the identification of the three phases of generative diffusion and the insights into manifold overfitting. While there are concerns about the linearity assumption and the real-world experimental evidence, the authors have made substantial improvements to the paper during the discussion phase, addressing many of these criticisms. The analysis has significant potential to inspire further research on the geometry of diffusion models and their generative properties.

Suggestions for Improvement Before Camera-Ready Submission:

Further refine the connections between sections to improve the paper's readability and logical flow.

Expand the discussion of how the insights from this paper could be used to improve training or evaluation of diffusion models in practice.

Include additional commentary on the limitations of the linear manifold assumption and future directions for extending the theory to nonlinear manifolds.

Conclusion:
This paper provides an innovative and rigorous theoretical framework for understanding the geometric properties of generative diffusion models. Despite some limitations, its contributions are significant and warrant publication. The constructive feedback from reviewers has been well-addressed, and the revised manuscript is a valuable addition to the field.

**Additional Comments On Reviewer Discussion:**

See above.

---

### Decision · Program_Chairs · 2025-01-22

Accept (Poster)